# Efficient Offline Policy Optimization with a Learned Model

**Zichen Liu**[†‡]  **Siyi Li**[†]  **Wee Sun Lee**[‡]  **Shuicheng Yan**[†]  **Zhongwen Xu**[†*]
[†]Sea AI Lab    [‡]National University of Singapore
{liuzc,xuzw}@sea.com   {zichen,leews}@comp.nus.edu.sg

## Abstract

MuZero Unplugged presents a promising approach for offline policy learning from logged data. It conducts Monte-Carlo Tree Search (MCTS) with a learned model and leverages Reanalyze algorithm to learn purely from offline data. For good performance, MCTS requires accurate learned models and a large number of simulations, thus costing huge computing time. This paper investigates a few hypotheses where MuZero Unplugged may not work well under the offline RL settings, including 1) learning with limited data coverage; 2) learning from offline data of stochastic environments; 3) improperly parameterized models given the offline data; 4) with a low compute budget. We propose to use a regularized one-step look-ahead approach to tackle the above issues. Instead of planning with the expensive MCTS, we use the learned model to construct an advantage estimation based on a one-step rollout. Policy improvements are towards the direction that maximizes the estimated advantage with regularization of the dataset. We conduct extensive empirical studies with BSuite environments to verify the hypotheses and then run our algorithm on the RL Unplugged Atari benchmark. Experimental results show that our proposed approach achieves stable performance even with an inaccurate learned model. On the large-scale Atari benchmark, the proposed method outperforms MuZero Unplugged by $43\%$. Most significantly, it uses only $5.6\%$ wall-clock time (i.e., 1 hour) compared to MuZero Unplugged (i.e., 17.8 hours) to achieve a $150\%$ IQM normalized score with the same hardware and software stacks. Our implementation is open-sourced at `https://github.com/sail-sg/rosmo`.

## 1 Introduction

Offline Reinforcement Learning (offline RL) (Levine et al., 2020) is aimed at learning highly rewarding policies exclusively from collected static experiences, without requiring the agent's interactions with the environment that may be costly or even unsafe. It significantly enlarges the application potential of reinforcement learning especially in domains like robotics and health care (Haarnoja et al., 2018; Gottesman et al., 2019), but is very challenging. By only relying on static datasets for value or policy learning, the agent in offline RL is prone to action-value over-estimation or improper extrapolation at out-of-distribution (OOD) regions. Previous works (Kumar et al., 2020; Wang et al., 2020; Siegel et al., 2020) address these issues by imposing specific value penalties or policy constraints, achieving encouraging results. Model-based reinforcement learning (MBRL) approaches have demonstrated effectiveness in offline RL problems (Kidambi et al., 2020; Yu et al., 2020; Schrittwieser et al., 2021). By modeling dynamics and planning, MBRL learns as much as possible from the data, and is generally more data-efficient than the model-free methods. We are especially interested in the state-of-the-art MBRL algorithm for offline RL, i.e., MuZero Unplugged (Schrittwieser et al., 2021), which is a simple extension of its online RL predecessor MuZero (Schrittwieser et al., 2020). MuZero Unplugged learns the dynamics and conducts Monte-Carlo Tree Search (MCTS) (Coulom, 2006; Kocsis & Szepesvári, 2006) planning with the learned model to improve the value and policy in a fully offline setting.

In this work, we first scrutinize the MuZero Unplugged algorithm by empirically validating hypotheses about when and how the MuZero Unplugged algorithm could fail in offline RL settings.

---

[*]Corresponding author.

The failures could also happen in online RL settings, but the intrinsic properties of offline RL magnify the effects. MCTS requires an accurate learned model to produce improved learning targets. However, in offline RL settings, learning an accurate model is inherently difficult especially when the data coverage is low. MuZero Unplugged is also not suitable to plan action sequences in stochastic environments. Moreover, MuZero Unplugged is a compute-intensive algorithm that leverages the compute power of an NVIDIA V100 $\times$ One week for running *each* Atari game (Schrittwieser et al., 2021). When trying to reduce the compute cost by limiting the search, MuZero Unplugged fails to learn when the number of simulations in tree search is low. Last but not least, the implementation of MuZero Unplugged is sophisticated and close-sourced, hampering its wide adoption in the research community and practitioners.

Based on the hypotheses and desiderata of MBRL algorithms for offline RL, we design ROSMO, a **R**egularized **O**ne-**S**tep **M**odel-based algorithm for **O**ffline reinforcement learning. Instead of conducting sophisticated planning like MCTS, ROSMO performs a simple yet effective one-step look-ahead with the learned model to construct an improved target for policy learning and acting. To avoid the policy being updated with the uncovered regions of the offline dataset, we impose a policy regularization based on the dataset transitions. We confirm the effectiveness of ROSMO first on BSuite environments by extensive experiments on the proposed hypotheses, demonstrating that ROSMO is more robust to model inaccuracy, poor data coverage, and learning with data of stochastic environments. We then compare ROSMO with state-of-the-art methods such as MuZero Unplugged (Schrittwieser et al., 2021), Critic Regularized Regression (Wang et al., 2020), Conservative Q-Learning (Kumar et al., 2020), and the vanilla Behavior Cloning baseline in both BSuite environments and the large-scale RL Unplugged Atari benchmark (Gulcehre et al., 2020). In the Atari benchmark, the ROSMO agent achieves a $194\%$ IQM normalized score compared to $151\%$ of MuZero Unplugged. It achieves this within a fraction of time (i.e., 1 hour) compared to MuZero Unplugged (i.e., 17.8 hours), showing an improvement of above $17\times$ in wall-clock time, with the same hardware and software stacks. We conclude that a high-performing and easy-to-understand MBRL algorithm for offline RL problems is feasible with low compute resources.

## 2 BACKGROUND

### 2.1 NOTATION

The RL problem is typically formulated with Markov Decision Process (MDP), represented by $\mathcal{M} = \{\mathcal{S}, \mathcal{A}, P, r, \gamma, \rho\}$, with the state and action spaces denoted by $\mathcal{S}$ and $\mathcal{A}$, the Markovian transition dynamics $P$, the bounded reward function $r$, the discount factor $\gamma$ and an initial state distribution $\rho$. At any time step, the RL agent in some state $s \in \mathcal{S}$ interacts with the MDP by executing an action $a \in \mathcal{A}$ according to a policy $\pi(a|s)$, arrives at the next state $s'$ and obtains a reward $r(s, a, s') \in \mathbb{R}$. The value function of a fixed policy and a starting state $s_0 = s \sim \rho$ is defined as the expected cumulative discounted rewards $V^\pi(s) = \mathbb{E}_\pi \left[ \sum_{t=0}^{\infty} \gamma^t r(s_t, a_t) \mid s_0 = s \right]$. An offline RL agent aims to learn a policy $\pi$ that maximizes $J_\pi(s) = V^\pi(s)$ solely based on the static dataset $\mathcal{D}$, which contains the interaction trajectories $\{\tau_i\}$ of one or more behavior policies $\pi_\beta$ with $\mathcal{M}$. The learning agent $\pi$ cannot have further interaction with the environment to collect more experiences.

### 2.2 OFFLINE POLICY IMPROVEMENT VIA LATENT DYNAMICS MODEL

Model-based RL methods (Sutton, 1991; Deisenroth & Rasmussen, 2011) are promising to solve the offline RL problem because they can effectively learn with more supervision signals from the limited static datasets. Among them, MuZero learns and plans with the latent model to achieve strong results in various domains (Schrittwieser et al., 2020). We next describe a general algorithmic framework extended from MuZero for the offline RL settings, which we refer to as *Latent Dynamics Model*.

Given a trajectory $\tau_i = \{o_1, a_1, r_1, \ldots, o_{T_i}, a_{T_i}, r_{T_i}\} \in \mathcal{D}$ and at any time step $t \in [1, T_i]$, we encode the observations into a latent state via the *representation* function $h_\theta$: $s_t^0 = h_\theta(o_t)$. We could then unroll the learned model recurrently using the *dynamics* function $g_\theta$ to obtain an imagined next state in the latent space and an estimated reward: $r_t^{k+1}, s_t^{k+1} = g_\theta(s_t^k, a_{t+k})$, where $k \in [0, K]$ denotes the imagination depth (the number of steps we unroll using the learned model $g_\theta$). Conditioned on the latent state the *prediction* function estimates the policy and value function:

$\pi_t^k, v_t^k = f_\theta\left(s_t^k\right)$. Note that we have two timescales here. The subscript denotes the time step $t$ in a trajectory, and the superscript denotes the unroll time step $k$ with the learned model.

In the learning phase, the neural network parameters are updated via gradient descent over the loss function as follows,

$$\ell_t(\theta) = \sum_{k=0}^{K} \ell^r\left(r_t^k, r_{t+k}^{\text{env}}\right) + \ell^v\left(v_t^k, z_{t+k}\right) + \ell^\pi\left(\pi_t^k, p_{t+k}\right) + \ell^{\text{reg}}\left(\theta\right), \tag{1}$$

where $\ell^r, \ell^v, \ell^\pi$ are loss functions for reward, value and policy respectively, and $\ell^{\text{reg}}$ can be any form of regularizer. The exact implementation for loss functions can be found in Appendix A.2. $r_{t+k}^{\text{env}}$ is the reward target from the environment (i.e., dataset), $z_{t+k}, p_{t+k} = \mathcal{I}_{(g,f)_{\theta'}}(s_{t+k})$ are the value and policy targets output from an improvement operator $\mathcal{I}$ using dynamics and prediction functions with target network parameters $\theta'$ (Mnih et al., 2013). Note that the predicted next state from the dynamics function is not supervised for reconstruction back to the input space. Instead, the model is trained implicitly so that policy and value predictions at time-step $t$ from imagined states at depth $k$ can match the improvement targets at real time-step $t + k$ from the environment. This is an instance of algorithms that apply the *value equivalence principle* (Grimm et al., 2020). Improvement operators $\mathcal{I}$ can have various forms; for example, MuZero Unplugged (Schrittwieser et al., 2021) applies Monte-Carlo Tree Search.

### 2.3 MONTE-CARLO TREE SEARCH FOR POLICY IMPROVEMENT

We briefly revisit Monte-Carlo Tree Search (MCTS) (Coulom, 2006; Kocsis & Szepesvári, 2006), which can serve as an improvement operator to obtain value and policy targets. To compute the targets for $\pi_t^k$ and $v_t^k$, we start from the root state $s_{t+k}^0$, and conduct MCTS simulations up to a budget $N$. Each simulation traverses the search tree by selecting actions using the pUCT rule (Rosin, 2011):

$$a^k = \arg\max_a \left[ Q(s,a) + \pi_{\text{prior}}(s,a) \cdot \frac{\sqrt{\sum_b n(s,b)}}{1 + n(s,a)} \cdot \left( c_1 + \log\left( \frac{\sum_b n(s,b) + c_2 + 1}{c_2} \right) \right) \right], \tag{2}$$

where $n(s,a)$ is the number of times the state-action pair has been visited during search, $Q(s,a)$ is the current estimate of the Q-value, $\pi_{\text{prior}}(s,a)$ is the probability of selecting action $a$ in state $s$ using the prior policy, and $c_1, c_2$ are constants. When the search reaches a leaf node $s^l$, it will expand the tree by unrolling the learned model $g_\theta$ with $a^l$ and appending a new node with model predictions $r_t^{l+1}, s_t^{l+1}, \pi_t^{l+1}, v_t^{l+1}$ to the search tree. Then the estimate of bootstrapped discounted return $G^k = \sum_{\tau=0}^{l-1-k} \gamma^\tau r^{k+1+\tau} + \gamma^{l-k} v^l$ for $k = l \ldots 0$ is backed up all the way to the root node, updating $Q$ and $n$ statistics along the path. After exhausting the simulation budget, the policy target $p_{t+k}$ is formed by the normalized visit counts at the root node, and the value target $z_{t+k}$ is the $n$-step discounted return bootstrapped from the estimated value at the root node[1]:

$$p_{\text{MCTS}}(a|s_t) = \frac{n(s_t^0, a)^{1/T}}{\sum_b n(s_t^0, b)^{1/T}},$$

$$z_{\text{MCTS}}(s_t) = \gamma^n \sum_a \left( \frac{n(s_{t+n}^0, a)}{\sum_b n(s_{t+n}^0, b)} \right) Q(s_{t+n}^0, a) + \sum_{t'=t}^{t+n-1} \gamma^{t'-t} r_{t'}^{\text{env}}. \tag{3}$$

### 3 METHODOLOGY

#### 3.1 MOTIVATION

With well learned function estimators $\{h_\theta, f_\theta, g_\theta\}$, planning with MCTS (Section 2.3) in the latent dynamics model (Section 2.2) has been shown to obtain strong policy and value improvements, and has been applied to offline RL (Schrittwieser et al., 2021) in *MuZero Unplugged*. However, the prohibitive computational power required by the search is limiting its practicability. For example,

---

[1] We re-index $t := t + k$ for an easier notation.

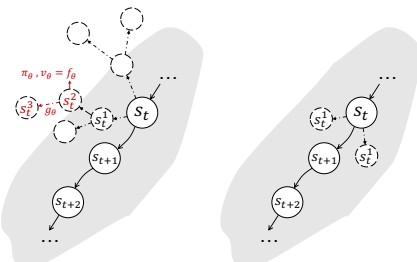

**Figure 1:** An illustration on the differences of Monte-Carlo Tree Search *(left)* and one-step look-ahead *(right)* for policy improvement. With limited offline data coverage, the search entering regions beyond the observed data may expand nodes on which $f_\theta$ fails to provide accurate estimations, and $g_\theta$ even leads to a worse region. These errors are compounded along the path as the search gets deeper, leading to detrimental improvement targets. One-step look-ahead is less likely to go outside the safe region. Illustrations are best viewed on screen.

experiments on the Atari benchmark (Bellemare et al., 2013) could take an NVIDIA V100 $\times$ One week to run a single game. Besides, in the offline RL settings, the state-action space coverage of the dataset is inherently limited, and further environment exploration is not possible. Thus, learned estimators may only be accurate in the safe regions covered by the dataset, and generalization outside the safe regions may lead to extrapolation error (Fujimoto et al., 2019). Even worse, the MCTS process unrolls the learned model recurrently with actions selected using Equation 2 (which depends on the estimation of $\pi_{\text{prior}}$), compounding the extrapolation errors along the search path.

Figure 1*(left)* illustrates when MCTS goes wrong as an improvement operator. Suppose we have encoded the observations through $h_\theta$ to get latent states $s_t, s_{t+1}, s_{t+2}, \dots$, which we refer to as observed nodes, as shown in the figure. The dotted edges and nodes in the left figure describe a simple MCTS search tree at node $s_t$, with simulation budget $N = 7$ and depth $d = 3$. We refer to the states unrolled by the learned model, $\{s_t^k\}$, as imagined nodes, with $k = 1, 2, 3$ denoting the depth of search. The regions far from the observed nodes and beyond the shaded area are considered unsafe regions, where a few or no data points were collected. Intuitively, policy and value predictions on imagined nodes in unsafe regions are likely erroneous, and so are the reward and next state imaginations. This makes the improvement targets obtained by Equation 3 unreliable. Furthermore, we also argue that MuZero Unplugged lacks proper regularization that constrains the policy from going too far away from the behavior policy to combat the distributional shift. In their algorithm, $\ell^{\text{reg}} = c||\theta||^2$ (in Equation 1) only imposes weight decay for learning stability.

---

**Algorithm 1** ROSMO

**Require:** dataset $\mathcal{D}$, initialized model parameters $\theta$
1: **while** True **do**
2:     Sample a batch of trajectory $\mathcal{B} \in \mathcal{D}$
3:     $z_t \leftarrow$ compute value target (Equation 8)     ▷
4:     $p_t \leftarrow$ compute policy target (Equation 6)     ▷
5:     $\ell^{\text{reg}} \leftarrow$ apply regularization (Equation 11)     ▷
6:     $\ell \leftarrow$ compute loss (Equation 1) on $\mathcal{B}$
7:     Update $\theta$ with gradient descent on $\ell$
8: **end while**

---

Motivated by the analysis above, the desiderata of a model-based offline RL algorithm are: compute efficiency, robustness to compounding extrapolation errors, and policy constraint. To addresses these desiderata, we design ROSMO, a **R**egularized **O**ne-**S**tep **M**odel-based algorithm for **O**ffline reinforcement learning based on value equivalence principle. As illustrated in Figure 1*(right)*, ROSMO performs one-step look-ahead to seek the improvement targets from a learned model, which is more efficient than MCTS and less affected by compounding errors. The overview of our complete algorithm is described in Algorithm 1. The algorithm follows Section 2.2 to encode the observations, unrolls the dynamics, makes predictions, and computes the loss. The blue triangles highlight our algorithmic designs on learning targets and regularization loss. We will derive them in the following sections.

## 3.2 A SIMPLE AND EFFICIENT IMPROVEMENT OPERATOR

In the algorithmic framework we outlined in Section 2.2, the improvement operator $\mathcal{I}$ is used to compute an improved policy and value target. Unfortunately, MuZero Unplugged uses the compute-

heavy and sophisticated MCTS to achieve this purpose, which is also prone to compounding extrapolation errors. In particular, the policy update of MuZero Unplugged is done by minimizing the cross entropy between the normalized visit counts and the parametric policy distribution:

$$\ell_{\mathrm{MCTS}}^{\pi} = -\sum_a \left( \frac{n(s,a)^{1/T}}{\sum_b n(s,b)^{1/T}} \log \pi(a|s) \right),$$ (4)

where the visit counts $n$ at the root node $s$ are summarized from the MCTS statistics.

We propose to learn the value equivalent model and use a more straightforward and much more efficient *one-step look-ahead* method to provide policy improvement. Specifically, our policy update is towards minimizing the cross entropy between a one-step (OS) improvement target and the parametric policy distribution:

$$\ell_{\mathrm{OS}}^{\pi} = -\mathbf{p}^{\mathsf{T}} \log \boldsymbol{\pi}.$$ (5)

The *policy target* $\mathbf{p}$ at state $s$ is estimated as:

$$p(a|s) = \frac{\pi_{\mathrm{prior}}(a|s) \exp\left(\mathrm{adv}_g(s,a)\right)}{Z(s)},$$ (6)

where $\pi_{\mathrm{prior}}$ is the prior policy (often realized by the target network $\pi_{\theta'}$), $\mathrm{adv}_g(s,a) = q_g(s,a) - v(s)$ is an approximation of the action advantage from the learned model $g_\theta$, and the factor $Z(s)$ ensures the policy target $\mathbf{p}$ is a valid probability distribution. The state value $v(s) = f_{\theta,v}(s)$ is from the prediction function conditioned on the current state. The action value $q_g(s,a)$ is estimated by unrolling the learned model one step into the future using dynamics function $g_\theta$ to predict the reward $r_g$ and next state $s'_g$, and then estimate the value at the imagined next state:

$$
\begin{aligned}
r_g, s'_g &= g_\theta(s,a), \\
q_g(s,a) &= r_g + \gamma f_{\theta,v}\left(s'_g\right).
\end{aligned}
$$ (7)

Intuitively, our policy target from Equation 6 adjusts the prior policy such that actions with positive advantages are favored, and those with negative advantages are discouraged.

Meanwhile, the *value target* $z$ is the $n$-step return bootstrapped from the value estimation:

$$z(s_t) = \gamma^n v_{t+n} + \sum_{t'=t}^{t+n-1} \gamma^{t'-t} r_{t'},$$ (8)

where $v_{t+n} = f_{\theta',v}(s_{t+n})$ is computed using the target network $\theta'$ and $r_{t'}$ is from the dataset. Compared to MuZero Unplugged, the value target is simpler, eliminating the dependency on the searched value at the node $n$ steps apart.

**Sampled policy improvement.** Computing the exact policy loss $\ell^p$ needs to simulate all actions to obtain a full $p(a|s)$ distribution, and then apply the cross entropy. It demands heavy computation for environments with a large action space. We thus sample the policy improvement by computing an estimate of $\ell^p$ on $N \leq |A|$ actions sampled from the prior policy, i.e., $a^{(i)} \sim \pi_{\mathrm{prior}}(s)$:

$$\ell_{\mathrm{OS}}^{\pi} \approx -\frac{1}{N} \sum_{i=1}^{N} \left[ \frac{\exp\left(\mathrm{adv}_g(s,a^{(i)})\right)}{Z(s)} \log \pi(a^{(i)}|s) \right].$$ (9)

The normalization factor $Z(s)$ for the $k$-th sample out of $N$ can be estimated (Hessel et al., 2021):

$$Z^{(k)}(s) = \frac{1 + \sum_{i \neq k}^{N} \exp\left(\mathrm{adv}_g(s,a^{(i)})\right)}{N}.$$ (10)

In this way, the policy update will not grow its computational cost along with the size of the action space.

### 3.3 BEHAVIOR REGULARIZATION FOR POLICY CONSTRAINT

Although the proposed policy improvement in Section 3.2 alleviates compounding errors by only taking a one-step look-ahead, it could still be problematic if this one step of model rollout leads

to an imagined state beyond the appropriate dataset extrapolation. Therefore, some form of policy constraint is desired to encourage the learned policy to stay close to the behavior policy.

To this end, we extend the regularization loss to $\ell^{\mathrm{reg}}(\theta) = c||\theta||^2 + \alpha\ell^{\mathrm{reg}}_{r,v,\pi}(\theta)$. The second term can be any regularization applied to reward, value, or policy predictions and is jointly minimized with other losses. While more sophisticated regularizers such as penalizing out-of-distribution actions' reward predictions could be designed, we present a simple behavior regularization on top of the policy $\pi$, leaving other possibilities for future research.

Our behavior regularization is similar to Siegel et al. (2020), but we do not learn an explicit behavior policy. Instead, we apply an advantage filtered regression directly on $\pi$ from the prediction function output:

$$\ell^{\mathrm{reg}}_{\pi}(\theta) = \mathbb{E}_{(s,a)\sim\mathcal{D}}\left[-\log\pi(a|s)\cdot H(\mathrm{adv}_g(s,a))\right], \tag{11}$$

where $H(x) = \mathbb{1}_{x>0}$ is the Heaviside step function. We can interpret this regularization objective as behavior cloning (maximizing the log probability) on a set of state-action pairs with high quality (advantage filtering).

### 3.4 SUMMARY

Our method presented above unrolls the learned model for one step to look ahead for an improvement direction and adjusts the current policy towards the improvement with behavior regularization. As illustrated in Figure 1, compared with MuZero Unplugged, our method could stay within the safe regions with higher probability and utilize the appropriate generalization to improve the policy. Our policy update can also be interpreted as approximately solving a regularized policy optimization problem, and the analysis can be found in Appendix B.

## 4 EXPERIMENT

In Figure 3, we have analyzed the deficiencies of MuZero Unplugged and introduced ROSMO as a simpler method to tackle the offline RL problem. In this section, we present empirical results to demonstrate the effectiveness and efficiency of our proposed algorithm. We firstly focus on the comparative analysis of ROSMO and MuZero Unplugged to justify our algorithm designs in Section 4.1, and then we compare ROSMO with existing offline RL methods and ablate our method in Section 4.2. Throughout this section we adopt the Interquartile Mean (IQM) metric (Agarwal et al., 2021) on the normalized score[2] to report the performance unless otherwise stated.

### 4.1 HYPOTHESIS VERIFICATION

To analyze the advantages of ROSMO over MuZero Unplugged, we put forward four hypotheses for investigation (listed in **bold**) and verify them on the BSuite benchmark (Osband et al., 2019). Similar to (Gulcehre et al., 2021), we use three environments from BSuite: catch, cartpole and mountain_car. However, we note that the released dataset by Gulcehre et al. (2021) is unsuitable for training model-based agents since it only contains unordered transitions instead of trajectories. Therefore, we generate an episodic dataset by recording experiences during the online agent training and use it in our experiments (see Appendix E for data collection details).

**(1) MuZero Unplugged fails to perform well in a low-data regime.** With a low data budget, the coverage of the state-action space shrinks as well as the safe regions (Figure 1). We hypothesize that MuZero Unplugged fails to perform well in the low-data regime since the MCTS could easily enter the unsafe regions and produce detrimental improvement targets. Figure 2(a) shows the IQM normalized score obtained by agents trained on sub-sampled datasets of different fractions. We can observe MuZero Unplugged degrades when the coverage becomes low and performs poorly with 1% fraction. In comparison, ROSMO works remarkably better in a low-data regime and outperforms MuZero Unplugged across all data coverage settings.

---

[2]Calculated as $x = (\mathrm{score} - \mathrm{score}_{\mathrm{random}})/(\mathrm{score}_{\mathbf{online}} - \mathrm{score}_{\mathrm{random}})$ per game. $x = 1$ means on par with the online agent used for data collection; $x = 1.5$ indicates its performance is $1.5\times$ of the data collection agent's.

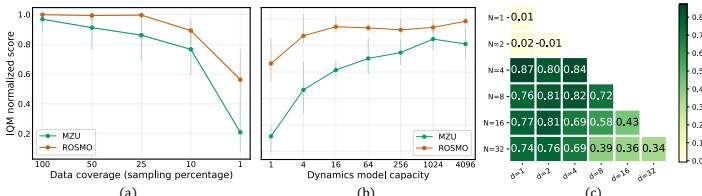

**Figure 2: (a)** IQM normalized score with different data coverage. **(b)** IQM normalized score with different dynamics model capacities. **(c)** IQM normalized score of MuZero Unplugged with different simulation budgets ($N$) and search depths ($d$).

**Table 1:** Comparison between ROSMO and MuZero Unplugged on catch with different noise levels ($\epsilon$).

| $\epsilon$ | MZU | ROSMO |
|---|---|---|
| 0 | $0.980_{\pm 0.060}$ | $1.0_{\pm 0.0}$ |
| 0.1 | $0.984_{\pm 0.054}$ | $1.0_{\pm 0.0}$ |
| 0.3 | $0.772_{\pm 0.253}$ | $0.900_{\pm 0.237}$ |
| 0.5 | $0.404_{\pm 0.422}$ | $0.916_{\pm 0.139}$ |

**(2) ROSMO is more robust in learning from stochastic transitions than MuZero Unplugged.** To evaluate the robustness of MuZero Unplugged and ROSMO in learning with data from stochastic environments, we inject noises during experience collection by replacing the agent's action with a random action for environment execution, with probability $\epsilon$. With the dataset dynamics being stochastic, MuZero Unplugged could fail to plan action sequences due to compounding errors. We hypothesize that ROSMO performs more robustly than MuZero Unplugged since ROSMO only uses a one-step look-ahead, thus has less compounding error. In Table 1, we compare the episode return of the two algorithms with the controlled noise level. The result shows that ROSMO is much less sensitive to the dataset noise and can learn robustly at different stochasticity levels.

**(3) MuZero Unplugged suffers from dynamics mis-parameterization while ROSMO is less affected.** The parameterization of the dynamics model is crucial for model-based algorithms. It is difficult to design a model with the expressive power that is appropriate for learning the dataset's MDP transition. The resulting under/over-fitting of the learned model may badly affect the performance of the overall algorithm. We hypothesize that MuZero Unplugged is more sensitive to the parameterization of dynamics than ROSMO. Figure 2(b) compares ROSMO with MuZero Unplugged for different dynamics model capacities trained on $10\%$ data. Since we use a multi-layer perceptron to model the dynamics function, the capacity is controlled by the number of hidden units. We show that MuZero Unplugged works best when the number of hidden units is $1024$, and its performance degrades significantly with less model capacity, likely due to the under-fitting of smaller networks. The effect of over-fitting is less obvious. In comparison, ROSMO performs stably with different dynamics model capacities and consistently outperform MuZero Unplugged in all settings.

**(4) MuZero Unplugged is sensitive to simulation budget and search depth.** Prior works have shown that the performance of MuZero agents declines with a decreasing simulation budget (Grill et al., 2020), and it is insensitive to search depth (Hamrick et al., 2021). Both works consider online RL settings, where new experience collection may correct prior wrong estimations. We hypothesize that in offline RL settings, the performance of MuZero Unplugged is sensitive to *both* simulation budget and search depth. In particular, a deeper search would compound extrapolation errors in offline settings, leading to harmful improvement targets. Figure 2(c) demonstrates the IQM normalized score of MuZero Unplugged with different simulation budgets and search depths. We can observe MuZero Unplugged fails to learn when $N$ is low, and it performs poorly when $N$ is high but with a deep search. This suggests that too low visit counts are not expressive, and too much planning may harm the performance, matching the findings in the online settings (Grill et al., 2020; Hamrick et al., 2021). Notably, limiting the search depth can ease the issue by a large amount, serving as further strong empirical evidence to support our hypothesis that deep search compounds errors, reinforcing our belief in the one-step look-ahead approach.

## 4.2 BENCHMARK RESULTS

After investigating what could go wrong with MuZero Unplugged and validating our hypotheses, we compare our method with other offline RL baselines on the BSuite benchmark as well as the larger-scale Atari benchmark with the *RL Unplugged* (Gulcehre et al., 2020) dataset.

**Baselines.** Behavior Cloning learns a maximum likelihood estimation of the policy mapping from the state space to the action space based on the observed data, disregarding the reward signal. Thus BC describes the average quality of the trajectories in the dataset and serves as a naive baseline. Conservative Q-Learning (CQL) (Kumar et al., 2020) learns lower-bounded action values by incor-

porating loss penalties on the values of out-of-distribution actions. Critic Regularized Regression (CRR) (Wang et al., 2020) approaches offline RL in a supervised learning paradigm and reweighs the behavior cloning loss via an advantage estimation from learned action values. CQL and CRR are representative offline RL algorithms with strong performance. MuZero Unplugged (MZU) (Schrittwieser et al., 2021) is a model-based method that utilizes MCTS to plan for learning as well as acting, and exhibits state-of-the-art performance on the RL Unplugged benchmark. MOReL (Kidambi et al., 2020) and MOPO (Yu et al., 2020) are another two model-based offline RL algorithms. MOReL proposes to learn a pessimistic MDP and then learn a policy within the learned MDP; MOPO models the dynamics uncertainty to penalize the reward of MDP and learns a policy on the MDP. Both of them focus on state-based control tasks and are not trivial to transfer to the image-based Atari tasks, hence are not compared here. Nevertheless, we do provide the details of our implementation for MOReL (Kidambi et al., 2020) and COMBO (Yu et al., 2021) (which extends MOPO (Yu et al., 2020)) and report the results in Appendix F.1.

**Implementation.** We use the same neural network architecture to implement all the algorithms and the same hardware to run all the experiments for a fair comparison. We closely follow MuZero Unplugged (Schrittwieser et al., 2021) to implement ROSMO and MuZero Unplugged, but use a down-scaled version for Atari to trade off the experimentation cost. We use the exact policy loss without sampling (Equation 5) for all ROSMO experiments in the main results and compare the performance of sampling in the ablation. For CRR and CQL we adapt the official codes for our experiments. For Atari we conduct experiments on a set of 12 Atari games due to limited computation resources[3]. We ensure they are representative and span the performance spectrum of MuZero Unplugged for fair comparison. More implementation details can be found in Appendix D.2.

| Method | Catch | MountainCar | Cartpole |
|--------|-------|-------------|----------|
| BC | $0.66_{\pm 0.02}$ | $-178.24_{\pm 43.12}$ | $589.78_{\pm 75.32}$ |
| CQL | $1.0_{\pm 0.0}$ | $-124.77_{\pm 30.95}$ | $416.48_{\pm 245.75}$ |
| CRR | $1.0_{\pm 0.0}$ | $-106.3_{\pm 15.99}$ | $997.11_{\pm 8.63}$ |
| MZU | $0.99_{\pm 0.01}$ | $-107.27_{\pm 3.84}$ | $890.64_{\pm 173.10}$ |
| ROSMO | $1.0_{\pm 0.0}$ | $-102.15_{\pm 3.04}$ | $990.68_{\pm 19.36}$ |

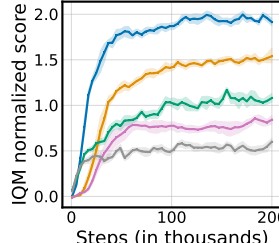 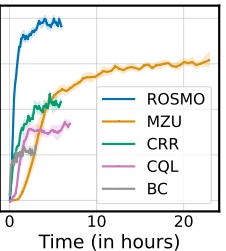

**Table 2:** BSuite benchmark results. Averaged episode returns measured at the end of training (200K steps) across 5 seeds.

**Figure 3:** Atari benchmark results. Aggregated IQM normalized score of different algorithms in terms of *(left)* sample efficiency and *(right)* wall-clock efficiency.

**Main results.** Table 2 shows the BSuite benchmark results of our algorithm and other baseline methods. ROSMO achieves the highest episode returns on catch and mountain_car with the lowest standard deviation. For cartpole, CRR performs slightly better than ROSMO, but we still observe ours outperforms the other baseline by a clear margin.

Figure 3*(left)* presents the learning curves with respect to the learner update steps, where it is clearly shown that ROSMO outperforms all the baselines and achieves the best learning efficiency. In terms of the final IQM normalized score, all offline RL algorithms outperform the behavior cloning baseline, suggesting that the reward signal is greatly helpful when the dataset contains trajectories with diverse quality. ROSMO obtains a 194% final IQM normalized score, outperforming MuZero Unplugged (151%), Critic Regularized Regression (105%), and Conservative Q-Learning (83.5%). Besides the highest final performance, we also note the ROSMO learns the most efficiently among all compared methods.

Figure 3*(right)* compares the wall-clock efficiency of different algorithms given the same hardware resources. Significantly, ROSMO uses only 5.6% wall-clock time compared to MuZero Unplugged to achieve a 150% IQM normalized score. With the lightweight one-step look-ahead design, the model-based ROSMO consumes similar learning time as model-free methods, widening its applicability to both offline RL researches and real-world applications.

---

[3]Even with our slimmed implementation (Appendix D.2 for details), a full run of all compared methods on the RL Unplugged benchmark (46 games) needs about 720 TPU-days for a single seed, which is approximately equivalent to an NVIDIA V100 GPU running for 2 years.

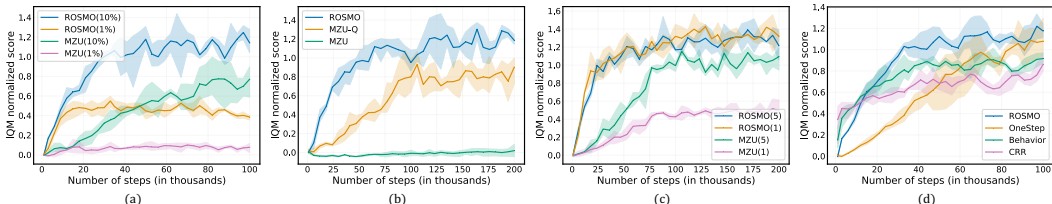

**Figure 4:** Learning curves of IQM normalized score on MsPacman. **(a)** Comparison of ROSMO and MuZero Unplugged in low data regime. **(b)** Comparison of ROSMO, MuZero Unplugged and MZU-Q when limiting the number of simulations (number of samples) to be $N = 4$. **(c)** Comparison of ROSMO and MuZero Unplugged when the model is unrolled with different steps for learning. **(d)** Ablation of the one-step policy improvement and the behavior regularization.

The results of individual games can be found in Appendix F.

**Ablations.** We present our ablation studies on data coverage, learning efficiency, model compounding errors, and decoupled ROSMO. Following the common practice (Schrittwieser et al., 2020; Hamrick et al., 2021), Ms. Pacman is chosen for the ablation studies.

**(a)** Figure 4(a) shows that ROSMO is able to outperform MuZero Unplugged in both 10% and 1% data regimes, replicating our hypothesis verification results on BSuite.

**(b)** To make MuZero Unplugged more compute-efficient and feasible, we could limit the number of simulations. However, prior works have shown that MuZero's policy target degenerates under low visit count (Grill et al., 2020; Hamrick et al., 2020). Hence, we also implement the MZU-Q variant which uses an MPO-style (Abdolmaleki et al., 2018) policy update, $\pi^{\mathrm{MPO}} \propto \pi_\theta \cdot \exp(Q^{\mathrm{MCTS}}/\tau)$, for a comprehensive comparison. Here $Q^{\mathrm{MCTS}}$ is the Q-values at the root node of the search tree, and $\tau$ is a temperature parameter set to be $0.1$ following Hamrick et al. (2021). Figure 4(b) shows that MZU fails to learn using $4$ simulations, while MZU-Q can somewhat alleviate the issues. Our sampled ROSMO performs well with a limited sampling budget.

**(c)** To alleviate the compounding errors (Janner et al., 2019), MuZero Unplugged unrolls the dynamics for multiple steps (5) and learns the policy, value, and reward predictions on the recurrently imagined latent state to match the real trajectory's improvement targets. It also involves complicated heuristics such as scaling the gradients at each unroll step to make the learning more stable. We ablate ROSMO and MuZero Unplugged using a single-step unroll for learning the model. Figure 4(c) shows that the performance of ROSMO is not sensitive to the number of unrolling (either $1$ or $5$), while MuZero Unplugged experiences a significant performance drop when only single-step unrolling is applied.

**(d)** We ablate the effect of behavior regularization. We compare our full algorithm ROSMO with two variants: OneStep - one-step policy improvement *without* regularization; Behavior - policy learning via standalone behavior regularization. Interestingly, the Behavior variant recovers the binary form of CRR (Wang et al., 2020), with an advantage estimated from the learned model. Figure 4(d) demonstrates that compared to OneStep, Behavior learns more efficiently at the early stage by mimicking the filtered behaviors of good quality, but is gradually saturated and surpassed by OneStep, which employs a more informative one-step look-ahead improvement target. However, OneStep alone without behavior regularization is not enough especially for low-data regime (see Appendix F.3 for more results). The full algorithm ROSMO combines the advantages from both parts to achieve the best learning results.

## 5 CONCLUSION

Starting from the analysis of MuZero Unplugged, we identified its deficiencies and hypothesized when and how the algorithm could fail. We then propose our method, ROSMO, a regularized one-step model-based algorithm for offline reinforcement learning. Compared to MuZero Unplugged, the algorithmic advantages of ROSMO are threefold: (1) it is computationally efficient, (2) it is robust to compounding extrapolation errors, and (3) it is appropriately regularized. The empirical investigation verified our hypotheses and the benchmark results demonstrate that our proposed algorithm can achieve state-of-the-art results with low experimentation cost. We hope our work will serve as a powerful and reproducible agent and motivate further research in model-based offline reinforcement learning.

## 6 ETHICS STATEMENT

This paper does not raise any ethical concerns. Our study does not involve human subjects. The datasets we collected do not contain any sensitive information and will be released. There are no potentially harmful insights or methodologies in this work.

## 7 REPRODUCIBILITY STATEMENT

To ensure the reproducibility of our experimental results, we included detailed pseudocode, implementation specifics, and the dataset collection procedure in the Appendix. More importantly, we released our codes as well as collected datasets for the research community to use, adapt and improve on our method.

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

# A  ALGORITHMIC DETAILS

## A.1  PSEUDOCODE

We present the detailed learning procedure of ROSMO in Algorithm 2. For notational convenience, we use a single slice of trajectory, but in practice we can take batches for parallel.

---

**Algorithm 2** ROSMO Pseudocode

---

**Require:** dataset $\mathcal{D}$, discount factor $\gamma$, initialized model parameters $\theta$, target parameters $\theta' = \theta$, unroll step $K$, TD step $n$, behavior regularization strenth $\alpha$, weight decay strength $c$, sampling budget $N$ (optional)

1: **while** True **do**
2:     Sample a trajectory $\tau_i \in \mathcal{D}$ with length $T_i$
3:     Sample a random time step $t \in [0, T_i - K - n - 1]$
4:     $s_t^0 \leftarrow h_\theta(o_t)$                                                 ▷ *representation* of root
5:     $\mathbf{r}_t, \mathbf{s}_t, \boldsymbol{\pi}_t, \mathbf{v}_t \leftarrow \text{UNROLL}(\theta, s_t^0, a_{t,...,t+K-1})$
6:     $\mathbf{p}_t, \mathbf{z}_t \leftarrow \text{IMPROVE}(\theta', o_{t,...,t+K+n}, r_{t,...,t+K+n}^{\text{env}})$
7:     $\ell^{\text{reg}} \leftarrow \alpha \text{BEHAVIORREGULARIZER}(\theta', \boldsymbol{\pi}_t, o_{t,...,t+K}, a_{t,...,t+K}) + c||\theta||$
8:     $\mathbf{r}^{\text{env}} \leftarrow$ directly get from $\tau_i$, indexed at $t+1, \ldots, t+K$
9:     $\ell^r, \ell^v, \ell^p \leftarrow$ compute losses following Appendix A.2
10:    Update $\theta$ with gradient descent on $\ell^r, \ell^v, \ell^p + \ell^{\text{reg}}$; update $\theta' = \theta$ with interval
11: **end while**

12: **function** UNROLL($\theta, s^0, a_{0,...,K-1}$)
13:     initialize vector containers $\mathbf{r}, \mathbf{s}, \boldsymbol{\pi}, \mathbf{v}$, with $\mathbf{r}^0 = 0, \mathbf{s}^0 = s^0$
14:     **for** $j = 0 \ldots K-1$ **do**
15:         $\boldsymbol{\pi}^j, \mathbf{v}^j \leftarrow f_\theta(\mathbf{s}^j)$                        ▷ *prediction* on root and the imaginary
16:         $\mathbf{r}^{j+1}, \mathbf{s}^{j+1} \leftarrow g_\theta(\mathbf{s}^j, a_j)$                   ▷ *dynamics*
17:     **end for**
18:     $\boldsymbol{\pi}^K, \mathbf{v}^K \leftarrow f_\theta(\mathbf{s}^K)$
19:     **return** $\mathbf{r}, \mathbf{s}, \boldsymbol{\pi}, \mathbf{v}$
20: **end function**

21: **function** IMPROVE($\theta', o_{0,...,K+n}, r_{0,...,K+n}^{\text{env}}$)
22:     initialize vector containers $\mathbf{p}, \mathbf{z}$ for policy and value targets
23:     $\mathbf{s} \leftarrow h_{\theta'}(o_{0,...,K+n})$
24:     $\boldsymbol{\pi}, \mathbf{v} \leftarrow f_{\theta'}(\mathbf{s})$
25:     **for** $j = 0 \ldots K$ **do**
26:         $\mathbf{adv} \leftarrow \text{ONESTEPLOOKAHEAD}(\theta', \mathbf{s}^j, \mathbf{v}^j)$
27:         $\mathbf{p}^j \leftarrow \boldsymbol{\pi} \exp(\mathbf{adv})/Z$
28:         $\mathbf{z}^j \leftarrow \gamma^n \mathbf{v}^{n+j} + \sum_{t'=j}^{j+n-1} \gamma^{t'-t} r_{t'}^{\text{env}}$
29:     **end for**
30:     **return** $\mathbf{p}, \mathbf{z}$
31: **end function**

32: **function** ONESTEPLOOKAHEAD($\theta', s, v$)
33:     $\mathbf{a} \leftarrow$ sample $N$ or enumerate actions
34:     $\mathbf{r}, \mathbf{s}' \leftarrow g_{\theta'}(s, \mathbf{a})$
35:     **return** $\mathbf{r} + \gamma f_{\theta', v}(\mathbf{s}') - v$
36: **end function**

37: **function** BEHAVIORREGULARIZER($\theta', \boldsymbol{\pi}, o_{0,...,K}, a_{0,...,K}$)
38:     $\mathbf{s} \leftarrow h_{\theta'}(o_{0,...,K+1})$
39:     $\mathbf{r}, \mathbf{s}' \leftarrow g_{\theta'}(\mathbf{s}, a_{0,...,K+1})$
40:     $\mathbf{adv} \leftarrow \mathbf{r} + \gamma f_{\theta', v}(\mathbf{s}') - f_{\theta', v}(\mathbf{s})$
41:     **return** $-\frac{1}{K+1} \sum_{j=0}^{K} \log(\boldsymbol{\pi}^j)^{\mathsf{T}} \mathbb{1}_{\mathbf{adv}^j > 0}$
42: **end function**

---

## A.2 TRAINING

An illustration of the training procedure can be found in Figure 5(left). To optimize the network weights, we apply gradient descent updates on $\theta$ over the loss defined in Equation 1. Specifically, our loss functions for policy, value and reward predictions are:

$$\ell^p(\boldsymbol{\pi}, \mathbf{p}) = -\mathbf{p}^\intercal \log \boldsymbol{\pi}, \tag{12}$$

$$\ell^v(\mathbf{v}, z') = -\phi(z')^\intercal \log \mathbf{v}, \tag{13}$$

$$\ell^r(\mathbf{r}, u') = -\phi(u')^\intercal \log \mathbf{r}, \tag{14}$$

where $z' = h(z), u' = h(r^{\text{env}})$ are the value and reward targets scaled by the invertible transform $h(x) = sign(x)(\sqrt{|x|+1} - 1 + \epsilon x)$, where $\epsilon = 0.001$ (Pohlen et al., 2018). We then apply a transformation $\phi$ to obtain the equivalent categorical representations of scalars, which then serve as the targets of cross entropy loss of the scalars' distribution predictions. For the policy prediction, the loss is its cross entropy with the improved policy target.

In Algorithm 2 (line-9), we have vectorized inputs of length $K + 1$ for the loss computation, thus for every element we apply the above loss functions and take the average.

We also follow MuZero (Schrittwieser et al., 2020) closely to scale the gradients at the start of dynamics function. However, we do not use the prioritized replay for simplicity and do not apply the normalization to the hidden states.

## B ANALYSIS

We can interpret minimizing the policy loss in Equation 12 as conducting a regularized policy optimization. Suppose our goal is to maximize the expected improvement $\eta(\pi) = J^\pi - J^\mu$ over the behavior policy $\mu(a|s) = \pi_\beta(a|s)$, which can be expressed in terms of the advantage $\text{adv}_\mu(s, a)$ with respect to $\mu$ (Kakade & Langford, 2002; Schulman et al., 2015):

$$\eta(\pi) = \mathbb{E}_{s \sim d_\pi(s)} \mathbb{E}_{a \sim \pi(a|s)} \left[ \text{adv}_\mu(s, a) \right], \tag{15}$$

where $d_\pi = \sum_{t=0}^\infty \gamma^t P(s_t = s | \pi)$ is the unnormalized discounted distribution of state visitation induced by policy $\pi$ (Sutton & Barto, 2020). In practice, we follow Schulman et al. (2015) to optimize an approximation $\hat{\eta}(\pi) = \mathbb{E}_{s \sim d_\mu(s)} \mathbb{E}_{a \sim \pi(a|s)} \left[ \text{adv}_\mu(s, a) \right]$, which provides a good estimate of $\eta(\pi)$ when $\pi$ and $\mu$ are close ($\leq \epsilon$) in terms of KL-divergence. Hence, we can use $\hat{\eta}$ as the surrogate objective and maximize it under a constraint, serving as a regularized policy optimization:

$$\arg\max_\pi \int_s d_\mu(s) \int_a \pi(a|s) \left[ \text{adv}_\mu(s, a) \right] da ds$$
$$\text{s.t.} \int_s d_\mu(s) \text{D}_{\text{KL}}(\pi(\cdot|s) | \mu(\cdot|s)) ds \leq \epsilon. \tag{16}$$

Using the Lagrangian of Equation 16, the optimal policy can be solved as:

$$\bar{\pi}(a|s) = \frac{1}{Z(s)} \mu(a|s) \exp\left( \text{adv}_\mu(s, a)/\beta \right), \tag{17}$$

where $Z(s)$ is the partition function, and $\beta$ is a Lagrange multiplier. The learning policy can be improved via projecting $\bar{\pi}$ back onto the manifold of parametric policies by minimizing their KL-divergence:

$$\arg\min_\pi \mathbb{E}_{s \sim \mathcal{D}} \left[ \text{D}_{\text{KL}} \left( \bar{\pi}(\cdot|s) || \pi(\cdot|s) \right) \right], \tag{18}$$

which is equivalent to minimizing a loss function over $\pi_\theta$:

$$\ell(\theta) = -\bar{\boldsymbol{\pi}}^\intercal \log \boldsymbol{\pi_\theta}. \tag{19}$$

Our one-step policy target in Equation 6 approximates $\bar{\pi}$ with $\beta$ fixed to 1 and $\pi_{\text{prior}}$ regularized towards the behavior policy $\mu$, yielding an approximate regularized policy improvement.

## C    MODEL-BASED OFFLINE REINFORCEMENT LEARNING

We discuss the related works in model-based offline reinforcement learning in this section. Model-based reinforcement learning refers to the class of methods that learn the dynamics function $P(s_{t+1}|s_t)$ and optionally the reward function $r(s, a, s')$, which are usually utilized for planning. Levine et al. (2020) has discussed several model-based offline RL algorithms in detail. The most relevant works to ours include MOReL (Kidambi et al., 2020), MOPO (Yu et al., 2020), COMBO (Yu et al., 2021) and MuZero Unplugged (Schrittwieser et al., 2021).

MOReL (Kidambi et al., 2020) proposes to learn an ensemble of dynamics models from the offline dataset, and then utilize it to construct a pessimistic-MDP (P-MDP), with which a normal RL agent can interact to collect experiences for learning. The construction of the P-MDP is based on the unknown state-action detector (USAD), which is realized by computing the ensemble discrepancy. If the discrepancy is larger than a threshold, then this state is treated as the absorbing state and taking the action will be given a negative reward as the penalty. This would help constrain the policy not entering the unsafe regions that is not covered by the dataset.

MOPO (Yu et al., 2020) takes a similar approach with MOReL by using uncertainty quantification to construct a lower bound for policy performance. The main difference is that a soft reward penalty is constructed by an estimate of the model's error and the policy is then trained in the resulting uncertainty-penalized MDP. COMBO (Yu et al., 2021) further extends MOPO by avoiding explicit uncertainty quantification for incorporating conservatism. Instead, a critic function is learned using both the offline dataset and the synthetic model rollouts, where the conservatism is achieved by extending CQL to penalize the value function in model simulated state-action tuples that are not in the support of the offline dataset.

MOReL, MOPO and COMBO have theoretically shown that the policy's performance under the learned model bounds its performance in the real MDP, and achieved promising empirical results on state-based benchmarks such as D4RL (Fu et al., 2020). However, it is unclear how such algorithmic frameworks can be transferred to image-based domains such as Atari in the RL Unplugged benchmark (Gulcehre et al., 2020). Unlike state-based environments, learning the dynamics model for image-based tasks is challenging, and using the ensemble of learned dynamics to help policy learning is even compute expensive, leading such algorithms not suitable for complex tasks.

Unlike methods discussed above, the *Latent Dynamics Model* (Section 2.2) does not aim to learn the environment dynamics explicitly and hence it does not require the reconstruction of the next state's observation, making it more practical for image-based tasks. Furthermore, instead of the two-stage process of learning the MDP and planning with the learned MDP, the latent dynamics model facilitates the end-to-end training for learning the model and using the model. Both MuZero Unplugged (Schrittwieser et al., 2021) and ROSMO (ours) lie in this family of algorithms. MuZero Unplugged relies on the Monte-Carlo Tree Search to plan with the learned model for proposing learning targets, which we have scrutinized in this paper and shown several deficiencies of (Section 3.1, Section 4.1). Motivated by the desiderata of model-based offline RL, including computational efficiency, robustness to compounding extrapolation errors and policy constraints, we developed ROSMO, which uses a one-step look-ahead for policy improvement and incorporates behavior regularization for policy constraint. Our simpler algorithm is computationally efficient, and able to outperform MuZero Unplugged as well as other offline RL methods (including model-free and model-based[4]) on the standard offline Atari benchmark (Section 4.2).

## D    EXPERIMENTAL DETAILS

### D.1    HARDWARE AND SOFTWARE

We use TPUv3-8 machines for all the experiments in Atari and use CPU servers with 60 cores for BSuite experiments. Our code is implemented using JAX (Bradbury et al., 2018).

---

[4]Two-stage model-based methods are compared in the Appendix F.1

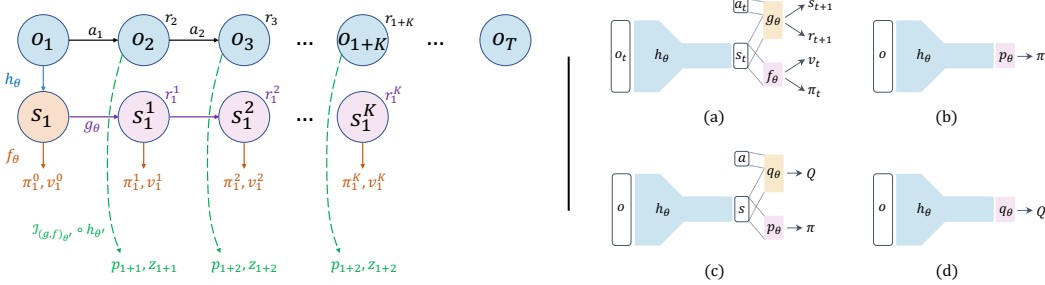

**Figure 5:** (*Left*) Illustration on the training procedure: root node unrolling, recurrently applying dynamics function, the model predictions and the improvement targets. (*Right*) Network architecture. **(a)** is for ROSMO and MuZero Unplugged; **(b)** is for Behavior Cloning; **(c)** is for Critic Regularized Regression; **(d)** is for Conservative Q-Learning.

### D.2 IMPLEMENTATION

#### D.2.1 NETWORK ARCHITECTURE

For both BSuite and Atari experiments, we use a network architecture based on the one used by MuZero Unplugged. In visual domains such as Atari games, the ResNet v2 style pre-activation residual blocks with layer normalization are used to model the representation, dynamics and prediction functions, while fully connected layers are used for simpler environments such as BSuite tasks.

We explain the network architecture for Atari in details. Figure 5(right) illustrates the network architectures used for the implementation of different algorithms. The blocks in light blue are the *representation* function (in the terminology of ROSMO or MuZero Unplugged), which encodes the observation into a hidden state. The overall network size is downscaled compared to Schrittwieser et al. (2021) due to the experimentation cost. For the stacked grayscale image input of size $84 \times 84 \times 4$, we firstly downsample as follows (with kernel size $3 \times 3$ for all convolutions):

- 1 convolution with stride 2 and 32 output channels.
- 1 residual block with 32 channels.
- 1 convolution with stride 2 and 64 output channels.
- 2 residual block with 64 channels.
- Average pooling with stride 2.
- 1 residual block with 64 channels.
- Average pooling with stride 2.

Then 6 residual blocks are used to complete the *representation* function. We use 2 residual blocks for the *dynamics* function (blocks in yellow) as well as the *prediction* function (blocks in pink). All residual blocks are with 64 hidden channels. All the network blocks are kept the same across different algorithms to ensure similar neural network capacity for a fair comparison.

For ROSMO and MuZero Unplugged, the input of the dynamics function is the latent state tiled with the one-hot encoded action vector. Two fully connected layers with 128 hidden units are used for the reward, value and policy predictions. The output size for policy is the size of action space, while the output size for reward and value is the number of bins (601) used for the categorical representation described in Appendix A.2.

For Behavior Cloning, we directly concatenate the representation function with the prediction function to model the policy network (Figure 5(right-b)). For Critic Regularized Regression, the Q network is similar to our dynamics function but without the next state prediction, and the policy network is based on our prediction function (Figure 5(right-c)). We follow the official code[5] for the choice of hyperparameters, and we also employ the same categorical representation for $Q$ value learning.

---

[5] https://github.com/deepmind/acme.

| Parameter | Value |
|---|---|
| Frames stacked | 4 |
| Sticky action | True |
| Discount factor | $0.997^4$ |
| Batch size | 512 |
| Optimizer | Adamw |
| Optimizer learning rate | $7 \times 10^{-4}$ |
| Optimizer weight decay | $10^{-4}$ |
| Learning rate decay rate | 0.1 |
| Max gradient norm | 5 |
| Target network update interval | 200 |
| Policy loss coefficient | 1 |
| Value loss coefficient | 0.25 |
| Unroll length | 5 |
| TD steps | 5 |
| Bin size | 601 |

**Table 3:** Atari hyperparameters shared by ROSMO and MuZero Unplugged.

| Parameter | Value |
|---|---|
| Representation MLP | [64, 64, 32] |
| Dynamics MLP | [32, 256, 32] |
| Prediction MLP | [32] |
| Discount factor | $0.997^4$ |
| Batch size | 128 |
| Optimizer | Adamw |
| Optimizer learning rate | $7 \times 10^{-4}$ |
| Optimizer weight decay | $10^{-4}$ |
| Learning rate decay rate | 0.1 |
| Max gradient norm | 5 |
| Target network update interval | 200 |
| Policy loss coefficient | 1 |
| Value loss coefficient | 0.25 |
| Unroll length | 5 |
| TD steps | 3 |
| Bin size | 20 |

**Table 4:** BSuite hyperparameters shared by ROSMO and MuZero Unplugged.

For Conservative Q-Learning, similarly, we follow the official code[6] and their hyperparameters, but with our network as illustrated in Figure 5(right-d).

### D.2.2 EXPERIMENT SETTINGS

The hyperparameters shared by ROSMO and MuZero Unplugged for Atari environments is given in Table 3, and that for BSuite environments is given in Table 4. In addition, the behavior regularization strength ($\alpha$) used in ROSMO is chosen to be 0.2. The simulation budget for MuZero Unplugged is 20 for Atari and 4 for BSuite, and the depth is not limited. We use the official library[7] to implement the MCTS used in MuZero Unplugged and its parameters follow the original settings (Schrittwieser et al., 2020).

For all experiments in Atari we use 3 seeds, and we use 5 seeds for all BSuite experiments. The comparison between ROSMO and MuZero Unplugged is made apple-to-apple by only replacing the policy and value targets as well as the regularizer.

---

[6] https://github.com/aviralkumar2907/CQL.
[7] https://github.com/deepmind/mctx

# E    BSUITE DATASET

We follow the setup of Gulcehre et al. (2021) to generate episodic trajectory data by training DQN agents for three tasks: cartpole, catch and mountain_car. We also add stochastic noise to the originally deterministic environments by randomly replacing the agent action with a uniformly sampled action with a probability of $\epsilon \in \{0, 0.1, 0.3, 0.5\}$. We use `envlogger`[8] to record complete episode trajectories through the training process. More details of the episodic dataset are provided in Table 5. We also record the score of a random policy and an online DQN agent on the three environments in Table 6, which can be used to normalize the episode return for evaluation.

| Environments | Number of episodes | Number of transitions | Average episode return |
|---|---|---|---|
| cartpole ($\epsilon = 0.0$) | 1,000 | 630,262 | 629.71 |
| cartpole ($\epsilon = 0.1$) | 1,000 | 779,491 | 779.01 |
| cartpole ($\epsilon = 0.3$) | 1,000 | 787,350 | 786.86 |
| cartpole ($\epsilon = 0.5$) | 1,000 | 527,528 | 526.75 |
| catch ($\epsilon = 0.0$) | 2,000 | 18,000 | 0.71 |
| catch ($\epsilon = 0.1$) | 2,000 | 18,000 | 0.60 |
| catch ($\epsilon = 0.3$) | 2,000 | 18,000 | 0.25 |
| catch ($\epsilon = 0.5$) | 2,000 | 18,000 | -0.04 |
| mountain_car ($\epsilon = 0.0$) | 500 | 82,342 | -164.68 |
| mountain_car ($\epsilon = 0.1$) | 500 | 147,116 | -294.23 |
| mountain_car ($\epsilon = 0.3$) | 500 | 138,262 | -276.52 |
| mountain_car ($\epsilon = 0.5$) | 500 | 167,688 | -335.37 |

**Table 5:** BSuite episodic dataset details.

| | random agent | online DQN agent |
|---|---|---|
| cartpole | 64.83 | 1,001.00 |
| catch | -0.66 | 1.00 |
| mountain_car | -1,000.00 | -102.16 |

**Table 6:** Episode return of random and online agent on BSuite environments.

# F    ADDITIONAL EMPIRICAL RESULTS

## F.1    BENCHMARK RESULTS FOR MODEL-BASED METHODS

In Section 4.2 we have compared ROSMO with Behavior Cloning, Conservative Q-Learning, Critic Regularized Regression and MuZero Unplugged on the offline Atari benchmark and presented the results in Figure 3. The comparison was made apple-to-apple as we standardized the neural network architecture and the optimization steps (Appendix D.2). However, comparing other model-based offline methods such as MOReL (Kidambi et al., 2020) and COMBO (Yu et al., 2021) in Figure 3 is less feasible. This is because these methods adopt a two-stage training procedure, where they first learn the MDP and then plan with the learned MDP. Moreover, MOReL and COMBO are not readily suitable for the Atari benchmark we use (MOReL only focuses on state-based tasks and COMBO's implementation and dataset are not released). Therefore, we implemented MOReL and COMBO and compared them with the other methods. For both of them, we have tuned the hyperparameters to report the results of the best configuration. In the following sections we introduce our implementation details and report our experimental results.

## F.1.1    DYNAMICS MODEL

Both MOReL and COMBO need to learn the dynamics model in the first stage. Since we are working with image-based tasks, we use the DreamerV2-style (Hafner et al., 2021) framework to learn

---

[8]https://github.com/deepmind/envlogger

the dynamics model. In particular, we only enable the world model learning in the DreamerV2, composed of the Recurrent State-Space Model, and the image, reward and discount predictor. We adapt the `pydreamer`[9] code base and follow the default hyper-parameters to train until convergence. We also trained an ensemble of dynamics models for uncertainty quantification.

### F.1.2  MOReL

Given the trained ensemble of dynamics models $\{f_1, f_2, \dots\}$, MOReL constructs a pessimistic MDP (P-MDP) by cross validating the collected trajectories in each learned model and computing the ensemble discrepancy as $\mathrm{disc}(s, a) = \max_{i,j} ||f_i(s, a) - f_j(s, a)||$. If $\mathrm{disc}(s, a)$ is larger than a certain threshold, the state $s$ is regarded as the terminal state and $r(s, a)$ is assigned to a small value as penalty. Since the MOReL framework is agnostic to the planner, we employ a PPO (Schulman et al., 2017) agent adapted from an open-source implementation[10] with strong online RL performance to learn from the P-MDP.

### F.1.3  COMBO

We refer to the implementation of a public repository `OfflineRL`[11] to adapt COMBO for Atari Games. We use the trained DreamerV2 to generate rollout data, where the dynamics model is randomly chosen from the ensemble for each rollout procedure. The rollout length is set to 10. The policy training part is very similar to CQL, where the only difference lies in that COMBO takes a mix of offline dataset and rollout dataset as input. The ratio between model rollouts and offline data is set to 0.5. The other hyperparameters just follow our CQL implementation.

### F.1.4  RESULTS

Following the setting in our ablation study, we choose MsPacman as a representative game to train both MOReL and COMBO with shared world models. We run the experiments with 3 different random seeds and report the mean and standard deviation. Table 7 shows that among all the model-based methods, ROSMO achieves the best result on MsPacman.

|  | MOReL | COMBO | MZU | ROSMO |
|---|---|---|---|---|
| MsPacman | $1476.478_{\pm 413.012}$ | $1538.33_{\pm 79.62}$ | $4539.048_{\pm 546.786}$ | $5019.762_{\pm 608.768}$ |

**Table 7:** Episode return on MsPacman of different model-based offline RL algorithms.

---

[9]`https://github.com/jurgisp/pydreamer`
[10]`https://github.com/vwxyzjn/cleanrl`
[11]`https://github.com/polixir/OfflineRL`

## F.2 LEARNING CURVES OF INDIVIDUAL ATARI GAMES

Figure 6 shows the learning curves in terms of IQM episode return for individual Atari games to compare ROSMO with other baseline methods, as complementary results for Figure 3. Table 8 records the numerical results of the IQM episode return of individual Atari games.

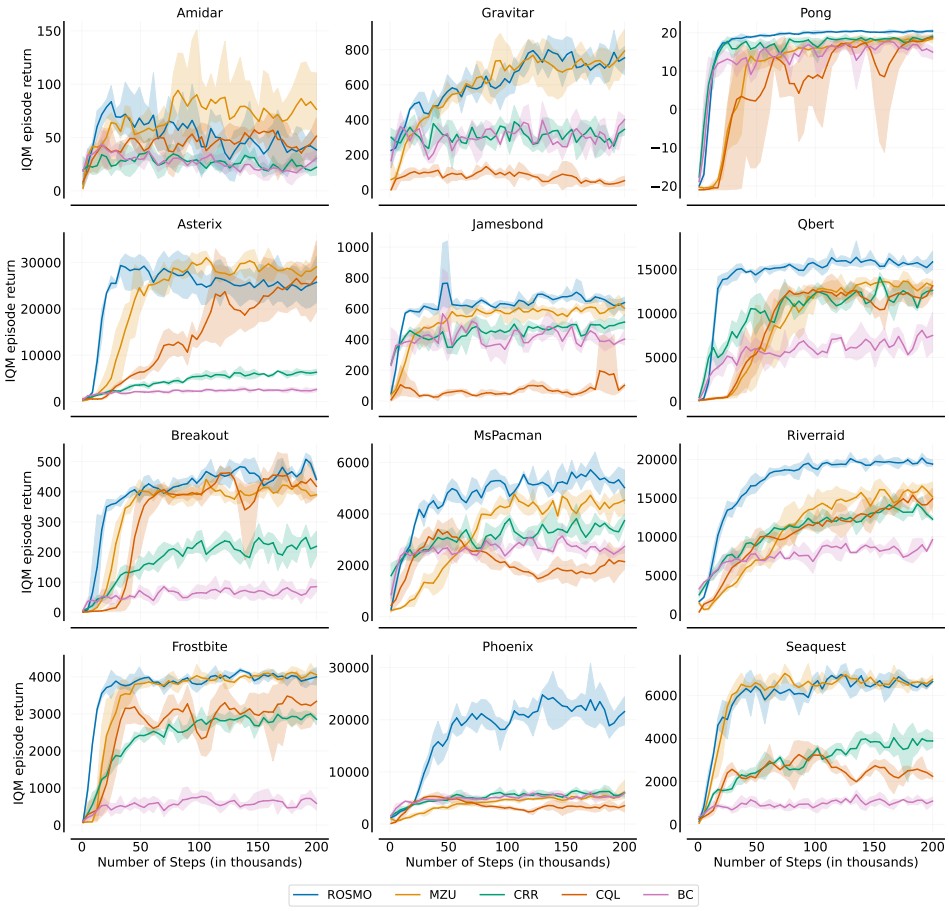

**Figure 6:** IQM episode return of different algorithms on individual Atari games.

|  | BC | CRR | CQL | MZU | ROSMO |
|---|---|---|---|---|---|
| Amidar | $30.381_{\pm 8.5}$ | $22.143_{\pm 9.143}$ | $51.286_{\pm 18.821}$ | $76.452_{\pm 31.107}$ | $38.405_{\pm 9.501}$ |
| Asterix | $2633.333_{\pm 392.857}$ | $6344.048_{\pm 698.214}$ | $26890.476_{\pm 8496.429}$ | $29061.905_{\pm 3667.857}$ | $25740.476_{\pm 3857.143}$ |
| Breakout | $85.143_{\pm 14.19}$ | $218.952_{\pm 40.571}$ | $418.238_{\pm 27.571}$ | $390.119_{\pm 4.357}$ | $440.905_{\pm 5.774}$ |
| Frostbite | $586.19_{\pm 145.357}$ | $2854.286_{\pm 113.571}$ | $3337.381_{\pm 544.643}$ | $4051.19_{\pm 107.5}$ | $3996.19_{\pm 223.006}$ |
| Gravitar | $400.0_{\pm 26.786}$ | $345.238_{\pm 50.0}$ | $52.381_{\pm 26.786}$ | $792.857_{\pm 108.929}$ | $753.571_{\pm 64.821}$ |
| Jamesbond | $402.381_{\pm 55.357}$ | $513.095_{\pm 46.429}$ | $102.381_{\pm 14.286}$ | $602.381_{\pm 40.878}$ | $639.286_{\pm 44.643}$ |
| MsPacman | $2733.333_{\pm 237.143}$ | $3736.905_{\pm 214.643}$ | $2141.667_{\pm 386.429}$ | $4539.048_{\pm 546.786}$ | $5019.762_{\pm 608.768}$ |
| Phoenix | $5901.905_{\pm 198.571}$ | $5954.286_{\pm 581.429}$ | $3510.238_{\pm 1066.429}$ | $6103.81_{\pm 1722.143}$ | $21550.476_{\pm 2689.643}$ |
| Pong | $14.976_{\pm 1.571}$ | $19.095_{\pm 0.571}$ | $18.762_{\pm 0.595}$ | $18.452_{\pm 1.107}$ | $20.452_{\pm 0.357}$ |
| Qbert | $7497.619_{\pm 2221.429}$ | $12618.452_{\pm 326.786}$ | $13114.286_{\pm 797.321}$ | $13121.429_{\pm 933.929}$ | $15848.81_{\pm 1049.107}$ |
| Riverraid | $9614.048_{\pm 528.929}$ | $12279.762_{\pm 280.357}$ | $14887.143_{\pm 1163.214}$ | $15156.905_{\pm 1965.714}$ | $19399.286_{\pm 407.143}$ |
| Seaquest | $1087.143_{\pm 169.286}$ | $3879.048_{\pm 421.905}$ | $2237.619_{\pm 150.714}$ | $6745.238_{\pm 170.0}$ | $6642.857_{\pm 87.857}$ |

**Table 8:** Numerical results of the IQM episode return of individual Atari games.

### F.3 COMPARISON ON ROSMO AND ONESTEP

Figure 7 shows the training curves of ROSMO and the OneStep variant (removing the behaviour regularization term defined in Equation 11). These results extend the ablation in Figure 4(d) with longer training time and more games. The comparison shows that ROSMO is able to achieve faster convergence, while resulting in similar or slightly better final performance. We further conducted experiments with only 1% data to verify the effect of behavior regularization when the data coverage is low. As shown in Figure 8, with limited data, ROSMO performs significantly better compared to OneStep. We can also observe that both methods suffer from over-fitting when the training goes longer due to limited sample size. To effectively handle this issue, we could resort to policy evaluation to early stop the training and select the best trained policy. Ideally in offline RL we need to use offline policy evaluation methods (Voloshin et al., 2021) for this purpose, which is unfortunately not trivial for difficult tasks. We leave this for future research since it is beyond the scope of this paper.

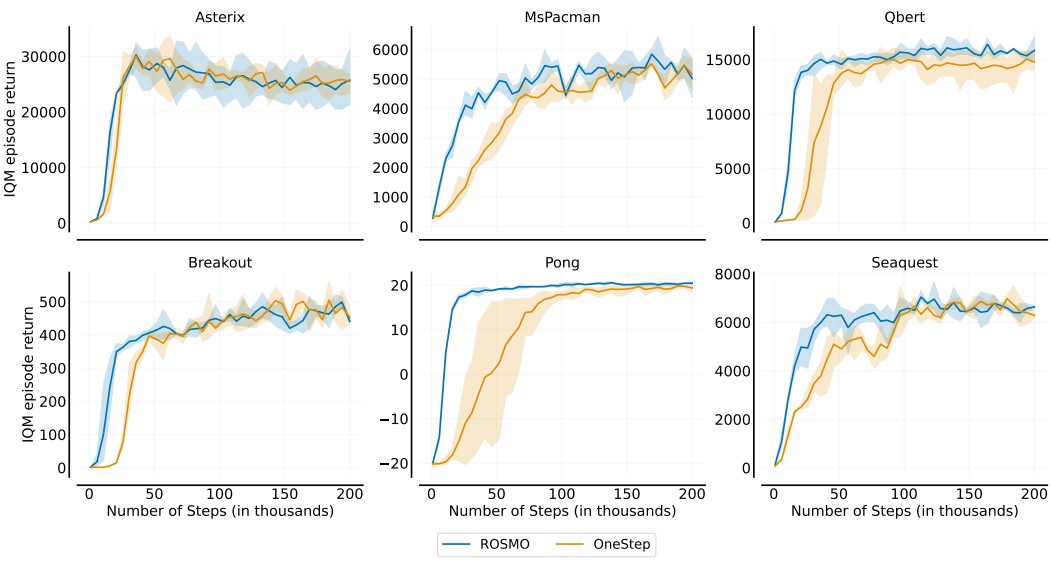

**Figure 7:** IQM episode return of ROSMO and OneStep.

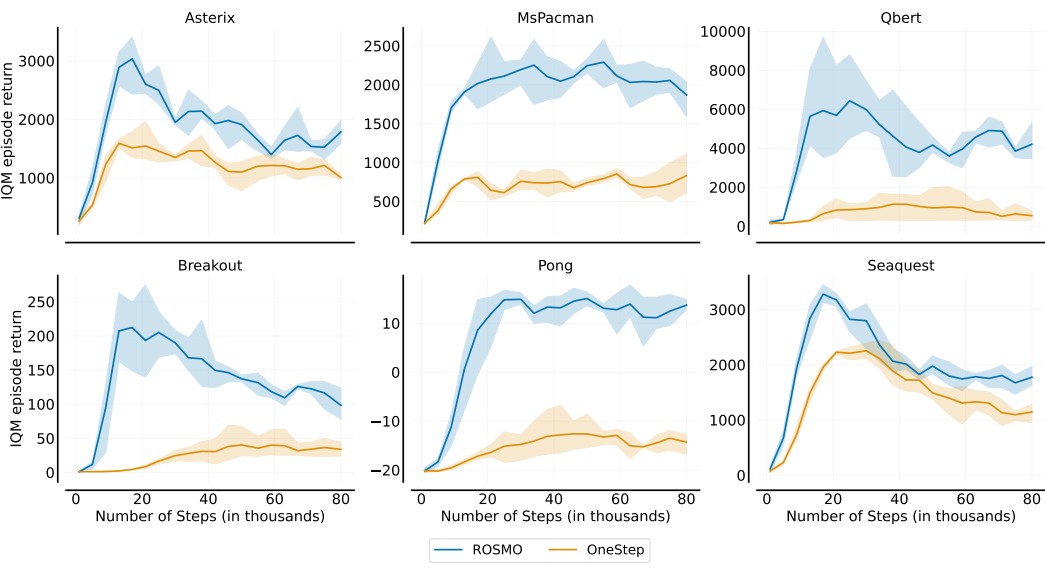

**Figure 8:** IQM episode return of ROSMO and OneStep trained only with 1% data.

### F.4 LEARNING CURVES FOR BSUITE EXPERIMENTS

Figure 9 and Figure 10 show the learning curves of individual settings in our BSuite analysis for noisy data model capacity.

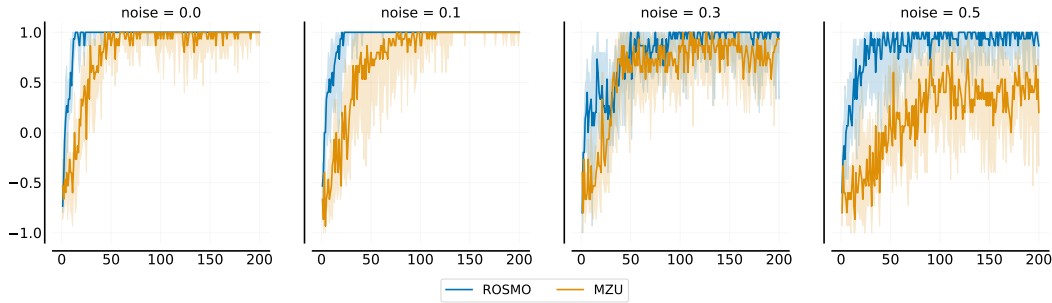

**Figure 9:** Learning curves of ROSMO and MuZero Unplugged on noisy catch environment as complementary results for Table 1.

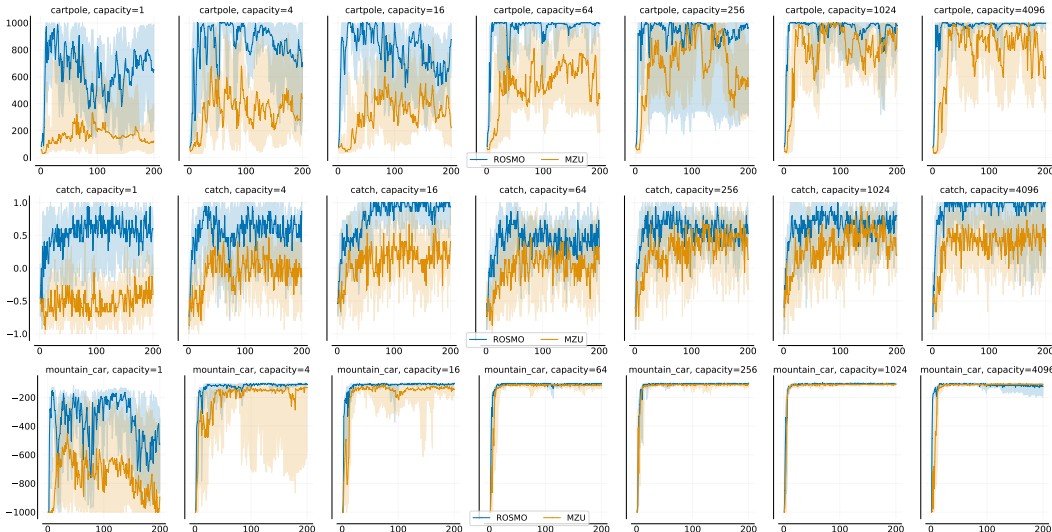

**Figure 10:** Learning curves of ROSMO and MuZero Unplugged on individual BSuite environment at different dynamics capacity, as complementary results for Figure 2(b).

