# OpenReview forum: "Efficient Offline Policy Optimization with a Learned Model"
_ICLR.cc/2023/Conference — ICLR 2023 poster_

### Official Review · Reviewer_tLyk · 2022-10-24

**Confidence:** 2
**Correctness:** 4
**Technical Novelty And Significance:** 3
**Empirical Novelty And Significance:** 3
**Recommendation:** 6

**Clarity, Quality, Novelty And Reproducibility:**

Clarity: this paper is clear.

Quality: technical details are provided, experimental details are also rather extensively documented.

Novelty: ideas used in this paper are pretty straightforward and are standard ingredients of offline RL (one-step lookahead is kind of analogous to Q-learning (?), behavior regularization also commonly used in past work),

Reproducibility: authors promised to open-source code upon publication.

**Strength And Weaknesses:**

**Strengths**
- Empirical results are pretty extensive and strong across the board.
- Sec 4.1 ablations help the reader understand what each modeling component addresses.

**Weaknesses**
- MOPO/MOReL: "both of them focus on state-based control tasks and hence are not compared here" - could you clarify what this mean?

**Summary Of The Paper:**

This paper studies offline model-based RL, specifically identified several shortcomings of the previously proposed MuZero Unplugged. The proposed ROSMO (Regularized One-Step Model-based Offline RL) incorporates (i) one-step lookahead instead of MCTS planning, (ii) behavior regularization.

Extensive experiments and ablation studies are conducted on the Atari benchmark and achieve impressive results, both in terms of performance and runtime efficiency.

**Summary Of The Review:**

Overall the paper is written quite clearly and empirical results are indeed strong, however my slight concern is where this contribution lies within the literature, perhaps a dedicated related works section would help - which should include more discussion on previous model-based offline RL approaches in "Offline RL Tutorial by Levine et al. 2020".

---

> ### Author Response · Authors · 2022-11-18
> **Response to Reviewer tLyk**
>
> We thank the reviewer for the helpful comments and questions that lead us to clarify the position of our work in other model-based methods. With the revision, we address your questions below.
>
> ***
> > **Q1**: MOPO/MOReL: "both of them focus on state-based control tasks and hence are not compared here" - could you clarify what this means?
>
> **A1**: MOPO [1] and MOReL [2] focus on low-dimensional state-based tasks (such as D4RL gym) to evaluate their algorithms. This is quite different from the image-based environments such as Atari that we are using. Building the world model ensemble for image-based tasks is computationally heavy and challenging if we follow their algorithmic frameworks. Using the learned ensemble to learn the policy makes the process complex and not scalable. Given that MOPO and MOReL not providing image-based evaluation and reproducing them requires many modifications to their algorithm, we did not compare them in our original submission. Nevertheless, as other reviewers required, we spent much effort adapting the MOReL and COMBO [3] (the extension of MOPO) algorithms to the image-based tasks and benchmarking them with ours. The results can be found in **Appendix F.1 on page 18**.
>
>
>
> In summary, we observe that among all compared methods (**MOReL**, **COMBO**, MuZero Unplugged, and ROSMO), ROSMO (**ours**) **achieves the best result**. Due to time and resource constraints, we only provide the results on MsPacman for this rebuttal round. We will extend the experiments to more games to validate this conclusion and update the results later. Additionally, we note that the two-stage ensemble-based methods (i.e., MOReL, COMBO) require even heavier computation for image-based tasks. For a single game, one must first train multiple dynamics models, then train the RL agent using the ensemble of the models (which also poses challenges to GPU memory). This might hinder the practicability of such methods. Model exploitation is also observed during training of **MOReL** and **COMBO**, where the learned model return continues to grow while the evaluation return starts to decrease.
>
> [1] Yu, T., Thomas, G., Yu, L., Ermon, S., Zou, J. Y., Levine, S., Finn, C. & Ma, T. (2020). MOPO: Model-based offline policy optimization. Advances in Neural Information Processing Systems, 33, 14129-14142.
>
> [2] Kidambi, R., Rajeswaran, A., Netrapalli, P., & Joachims, T. (2020). MOReL: Model-based offline reinforcement learning. Advances in neural information processing systems, 33, 21810-21823.
>
> [3] Yu, T., Kumar, A., Rafailov, R., Rajeswaran, A., Levine, S., & Finn, C. (2021). COMBO: Conservative offline model-based policy optimization. Advances in neural information processing systems, 34, 28954-28967.
>
> ***
> > **Q2**: more discussion on previous model-based offline RL approaches in "Offline RL Tutorial by Levine et al. 2020".
>
> **A2**: We have added a section in **Appendix C on p15** discussing the connections of our method with other model-based offline methods.

---

> ### Author Response · Authors · 2022-11-24
> **any further questions?**
>
> Dear Reviewer tLyk,
>
> We have addressed your concerns about the clarity of the baseline algorithm description. Also, two algorithms are added for comparison. Finally, discussions in a broader group of offline RL literature have been added as well. If you have any further questions, please let us know. If you are pleased with the current version, could you please consider raising your evaluation to a strong accept?
>
> Thanks,
> Authors

---

### Official Review · Reviewer_Rze2 · 2022-10-25

**Confidence:** 4
**Correctness:** 4
**Technical Novelty And Significance:** 3
**Empirical Novelty And Significance:** 3
**Recommendation:** 8

**Clarity, Quality, Novelty And Reproducibility:**

The author provide a detailed pseudo code in the appendix and will release their code if the paper is accepted.

Question:
- In the pseudo code, the author use K as the number of unroll steps. This K is only used for the ablation study to compare the performance between K=1 and K=5 and the ROSMO paper is using K=1 by default hence the "one-step" update? Is that correct? Same question for the loss in equation (1), ROSMO is using K=1, it is just for the ablation that K=5 is used?

- p5: v_t+n = f_theta'(s_t+n), I think it should be f_theta', v?
- p5: pi_prior is the prior policy (often realized by the target network). In practice in the paper I assume it is always the target network that is used for the prior policy?
- p7: Muzero is insensitive to search depth (Hamrick et al.) : I think the conclusion is a bit more nuanced than this. They say that "simple and shallow forms of planning **may** be sufficient". Moreover "Indeed, out of all our environments, only Acrobot and 9x9 Go strongly benefited from search at evaluation time. We therefore emphasize that for work which aims to build flexible and generalizable model-based agents, it is important to evaluate on a diverse range of settings that stress different types of reasoning"
- p7 (last paragraph): Figure 2(b) -> Figure 2(c)

**Strength And Weaknesses:**

The paper is well written and well organized. It was easy to read and understand. I liked that the authors recalls the ideas behind Muzero Unplugged in section 2.2 (in a more general way) and the MCTS policy improvement used by Muzero Unplugged in section 2.3, I think this eases the reading of the paper for people not very familiar with Muzero Unplugged.

Muzero Unplugged is a not so simple algorithm to implement and has no official open sourced implementation as far as I know. I agree with the authors write that "Muzero Unplugged is sophisticated". Having a simpler, yet as effective (and even better) algorithm is a great contribution for the community, especially if the implementation is to be made open sources after acceptance of the paper.

Empirical evaluations of Muzero Unplugged and ROSMO as well as the ablation study show the pertinence and the superiority of the ROSMO algorithm.

It would be better to say that having a one-step update might not be the only way to solve the compounding of errors as well as the fact that Muzero can go in unsafe regions. For instance having a model with its own uncertainty could be another direction to avoid unsafe regions and planning too much into the future. I think that planning (with MCTS or a different search algorithm) could make better targets if the uncertainty is used correctly. I agree that this would make a more complicated algorithm and for that it's good to have the ROSMO algorithm as it is simpler and will provide a good baseline when developing other algorithms.

The part on the stochastic transitions in section 4.1 is the least clear for me: why not add noise in the observations? why is it connected to the compounding of errors?

Designing simpler algorithms is a nice contribution to the community and for this reason I would be in favor of accepting the paper.

It is a pity that the authors are not using the D4RL benchmark which has become the default benchmark for offline RL and for which performances of MOPO/CQP/Morel etc... are available. It would have been nice to compare the ROSMO algorithm to these algorithms. Muzero unplugged is said to be the state of the art but was not compared to these algorithms to the best of my knowledge which are uncertainty based.

**Summary Of The Paper:**

The paper introduces ROSMO, a simpler and better version of Muzero Unplugged for offline policy optimization. In the offline setting, Muzero Unplugged can suffer from limited data-coverage, inaccurate models and an expensive compute budget. Instead of relying on an expensive MCTS and suffering from out of distribution errors as well as compounding of errors, ROSMO resorts to a one step lookahead approach to learn the same components as the one needed by Muzero Unplugged (policy and value targets) and a behavior cloning regularization to prevent going into an unsafe regions. ROSMO outperforms Muzero Unplugged on the Bsuite and Atari benchmarks. The authors also provide an ablation study.

**Summary Of The Review:**

Overall I think this is a nice paper and good contribution to the community. A comparison to other model-based algorithms for offline RL on the D4RL would have been a big plus (Morel/MOPO, ...).

---

> ### Author Response · Authors · 2022-11-18
> **Response to Reviewer Rze2 (2/n)**
>
> > **Q4**: p5: v_t+n = f_theta'(s_t+n), I think it should be f_theta', v?
>
> **A4**: Yes, thanks for pointing it out, and we have fixed it.
>
>
> ***
> > **Q5**: p5: pi_prior is the prior policy (often realized by the target network). In practice in the paper I assume it is always the target network that is used for the prior policy?
>
> **A5**: Yes, we maintain a copy of parameters that is 200 steps older and use it to infer the prior policy.
>
> ***
> > **Q6**: p7: Muzero is insensitive to search depth (Hamrick et al.) : I think the conclusion is a bit more nuanced than this. They say that "simple and shallow forms of planning may be sufficient". Moreover "Indeed, out of all our environments, only Acrobot and 9x9 Go strongly benefited from search at evaluation time. We therefore emphasize that for work which aims to build flexible and generalizable model-based agents, it is important to evaluate on a diverse range of settings that stress different types of reasoning"
>
> **A6**: In Hamrick et al. the empirical results suggest that shallow search is enough for most cases in the online RL setting, which aligns with our findings in the offline setting (see Figure 2(c), $N=4, d=1$ yields the best result). In fact, this is also one of the motivations for us to employ a shot horizon model rollout in our algorithm. We believe there should be more interesting or effective ways to design the improvement operator other than MCTS (in MuZero Unplugged) or OneStep (in ours) for offline model-based RL, and we leave it for future research.
>
> > **Q7**: p7 (last paragraph): Figure 2(b) -> Figure 2(c)
>
> **A7**: Thanks for the correction, and we have fixed it.

---

> > ### Comment · Reviewer_Rze2 · 2022-12-02
> > **Thank you for the detailled response and the additional results**
> >
> > Sorry for the delay. Thank you very much for adding results with Morel. I understand that algorithms such as Morel can be hard to apply on image-based tasks and I appreciate the effort done by the authors to compare them on Ms Pacman. Another solution would have been to run Muzero and ROSMO on the D4RL Mujoco non-image based tasks to see how ROSMO performs there compared to uncertainty based algorithms. I am not very familiar with the Bsuite benchmark but it seems that the environments selected by the authors are not imaged-based? In this case why did you choose this benchmark instead of the more common D4RL one used by Morel/Mopo/Combo/CQL?

---

> > > ### Author Response · Authors · 2022-12-03
> > > **Response to the follow-up question**
> > >
> > > Thank you for the feedback and follow-up questions! Applying MuZero to continuous domain tasks such as D4RL contains is not trivial because Monte-Carlo Tree Search could not operate in the continuous space. Another dedicated work [1] proposes `Sampled MuZero` for continuous space. However, the official implementation [2] does not open-source it. Hence it is not easy to make a fair comparison (our discrete MuZero is based on this official implementation for a fair comparison). Based on these reasons, we choose to stay with the discrete action space for our experiments to compare ROSMO with MuZero.
> > >
> > > Environments from BSuite also contain discrete action space, but they are more lightweight and principled, and thus they are chosen to perform comparative experiments to validate our algorithm designs. Our paper uses BSuite to validate our hypothesis on how and when MuZero Unplugged may not work. BSuite, proposed by DeepMind, is a collection of carefully-designed experiments that investigate the core capabilities of reinforcement learning (RL) agents. Please refer to their paper for details https://openreview.net/forum?id=rygf-kSYwH. BSuite has been used by Offline RL literature [3] for the understanding of offline RL algorithms.
> > >
> > > In the future, we plan to extend our work to continuous domain tasks such as the ones in D4RL.
> > >
> > >
> > > [1] Hubert, T., Schrittwieser, J., Antonoglou, I., Barekatain, M., Schmitt, S., & Silver, D. (2021, July). Learning and planning in complex action spaces. In International Conference on Machine Learning (pp. 4476-4486). PMLR.
> > >
> > > [2] https://github.com/deepmind/mctx
> > >
> > > [3] Gulcehre, C., Colmenarejo, S. G., Wang, Z., Sygnowski, J., Paine, T., Zolna, K., ... & de Freitas, N. (2021). Regularized behavior value estimation. arXiv preprint arXiv:2103.09575.

---

> > > > ### Comment · Reviewer_Rze2 · 2022-12-04
> > > > **Thanks again**
> > > >
> > > > Thank you for the response. Could you add a sentence in the current version explaining why you use Bsuite instead of D4RL ?
> > > >
> > > > I think this paper provides a valuable algorithm for the community and should be accepted. I will increase my score.

---

> > > > > ### Author Response · Authors · 2022-12-05
> > > > > **Will add a sentence on why bsuite over D4RL**
> > > > >
> > > > > Dear Reviewer Rze2,
> > > > >
> > > > > Thanks a lot for your appreciation of our work!
> > > > >
> > > > > Sure, we will add a sentence to explain why we chose Bsuite instead of D4RL in the revision. The edit is not open, but we will do it in the final revision.

---

> ### Author Response · Authors · 2022-11-18
> **Response to Reviewer Rze2 (1/n)**
>
> We thank the reviewer for the detailed and thoughtful comments, which helped us to refine the paper. With the revision, we address your questions below.
>
> ***
> > **Q1.1**: The part on the stochastic transitions in section 4.1 is the least clear for me: why not add noise in the observations?
>
> **A1.1**: We agree that adding noise to the observations is another way to make the stored transitions stochastic. In our experiment, we follow RBVE [1] to randomly replace the action the data collecting agent issued for real execution in the environment but still record the originally issued action. This way, the recorded transitions $(s, a, s’)$ would encode stochasticity. Suppose in the original MDP, $s_1 = P(s_0, a_0)$; now the stochastic transitions encode $P^\prime$ such that $P^\prime(s_0, a_0)$ can lead to some state $s_2 = P(s_0, a_1)$ with randomly selected $a_1$. This would mimic the stochastic environment and pose challenges for model learning.
>
> [1] Gulcehre, C., Colmenarejo, S. G., Wang, Z., Sygnowski, J., Paine, T., Zolna, K., ... & de Freitas, N. (2021). Regularized behavior value estimation. arXiv preprint arXiv:2103.09575.
>
> ***
> > **Q1.2**: why is it connected to the compounding of errors?
>
> **A1.2**: Since the dynamics model is deterministic, the stochastic transitions would lead to inaccurately learned models. Using the model in MuZero involves planning with tree search, where each step could introduce errors with the inaccurate model. In stochastic environments, the action sequence needs to depend on occurred stochastic events, which deterministic models could not capture. Hence the planning can be confounded and lead to wrong estimations. The latest work along the MuZero line [1] incorporates a stochastic model and stochastic tree search, which intends to solve this issue. ROSMO only employs a one-step look-ahead and alleviates compounding errors, offering better improvement targets.
>
>
> [1] Antonoglou, I., Schrittwieser, J., Ozair, S., Hubert, T. K., & Silver, D. (2022). Planning in Stochastic Environments with a Learned Model. In International Conference on Learning Representations.
>
> ***
> > **Q2**: It would have been nice to compare the ROSMO algorithm to uncertainty-based algorithms.
>
> **A2**: Thanks for the suggestion. To our knowledge, other model-based methods that utilize uncertainty mainly focus on low-dimensional state-based tasks. It is unclear how their performance would transfer to more complex image-based domains, such as Atari, that our work is based on. We have spent a lot of effort (**two human-week + 8 A100 machines**) in implementing MOReL and tested it in the Atari MsPacman environment (which is a representative game in Atari benchmarks, as explained in the paper). Specifically, we adapted DreamerV2 to train an ensemble of world models, followed by the MOReL algorithm to construct a pessimistic MDP using the ensemble. The implementation details and results of MoReL can be found in **Appendix F.1 on page 18**. We also attached all our codes (not well-cleaned yet working) for our original experiments and the newly added MOReL experiments.
>
>
> In summary, we observe that among all compared methods (**MOReL**, **COMBO**, MuZero Unplugged, and ROSMO), ROSMO (**ours**) **achieves the best result**. Due to time and resource constraints, we only provide the results on MsPacman for this rebuttal round. We will extend the experiments to more games to validate this conclusion and update the results later. Additionally, we note that the two-stage ensemble-based methods (i.e., MOReL, COMBO) require even heavier computation for image-based tasks. For a single game, one must first train multiple dynamics models, then train the RL agent using the ensemble of the models (which also poses challenges to GPU memory). This might hinder the practicability of such methods. Model exploitation is also observed during training of **MOReL** and **COMBO**, where the learned model return continues to grow while the evaluation return starts to decrease.
>
> ***
> > **Q3**: In the pseudo code, the author use K as the number of unroll steps. This K is only used for the ablation study to compare the performance between K=1 and K=5 and the ROSMO paper is using K=1 by default hence the "one-step" update? Is that correct? Same question for the loss in equation (1), ROSMO is using K=1, it is just for the ablation that K=5 is used?
>
> **A3**: In ROSMO we train with $K=5$. This is to ensure the training settings are as similar as possible to MZU’s. We hypothesize that such multi-step unroll for training is necessary for MZU to reduce compounding errors, thus helping the MCTS for improvement. Hence, we ablate this factor by setting $K=1$ for MZU and ROSMO and validating our hypothesis.

---

> ### Author Response · Authors · 2022-11-24
> **any further questions?**
>
> Dear Reviewer Rze2,
>
> We have added the suggested baseline algorithm with uncertainty estimation, i.e., MOReL, into our revision. As a result, our method ROSMO outperforms the newly added baseline.
>
> We would like to know if you have further concerns/questions so we can elaborate more. We believe that the current improved version is stronger than the initial draft. Could you please consider raising the evaluation score to a strong accept?
>
> Thanks,
> Authors

---

### Official Review · Reviewer_Dxeb · 2022-10-31

**Confidence:** 4
**Correctness:** 3
**Technical Novelty And Significance:** 2
**Empirical Novelty And Significance:** 3
**Recommendation:** 5

**Clarity, Quality, Novelty And Reproducibility:**

Overall, the paper is well-written and easy to follow. The technical novelty seems to be limited, given that each component of the improvements to MZU is not new.

**Strength And Weaknesses:**

[Strengths]
1. The paper is well-written and easy to follow. The modification to MZU for offline RL is well-motivated, and it makes the algorithm simpler yet more effective in terms of computation cost and performance.
2. ROSMO outperforms baseline algorithms in the experiments.


[Weaknesses]
1. Technical novelty is a bit limited. The short-horizon rollout for MBRL has already been widely used (e.g., [1]), and the behavior regularization is also very similar to the one in previous works (e.g. [2,3]). I am curious to see the result when only (11) is used for the policy improvement while not using (9), since it seems (11) can also serve as a policy improvement.
2. There are missing comparison with model-based offline RL algorithms in the experiments, e.g. [4,5,6].
3. In (1), there is no explicit definition of each term, i.e. $\ell^r$, $\ell^v$, and $\ell^p$.
4. In Figure 4d, it seems OneStep is not yet converged. It would be great to provide the result with a longer number of steps to see whether ROSMO clearly outperforms OneStep at convergence. In the current result of Figure 4d, the conclusion is unclear to me.
5. It would be great to include the result of baseline algorithms in Figure 2. Why is ROSMO only being compared with MZU?
6. In section 4.1.(3), why is MZU more sensitive to the parametrization of dynamics? It would be great to provide some possible reasons.


[1] Janner et al., When to Trust Your Model: Model-Based Policy Optimization, 2019

[2] Oh et al., Self-Imitation Learning, 2018

[3] Wang et al., Critic Regularized Regression, 2020

[4] Yu et al., COMBO: Conservative Offline Model-Based Policy Optimization, 2021

[5] Kidambi et al., MOReL : Model-Based Offline Reinforcement Learning, 2020

[6] Yu et al., MOPO: Model-based Offline Policy Optimization, 2020



**Summary Of The Paper:**

This paper presents ROSMO, a model-based offline RL algorithm based on MuZero Unplugged (MZU). ROSMO improves MZU in the offline RL setting in two main ways. First, it only performs one-step rollout for advantage estimation, rather than MCTS with multi-step rollouts. Second, it uses advantage-based policy improvement with advantage-filtered behavior cloning. ROSMO outperforms MZU in a low-data regime and stochastic transition and is less sensitive to network parameters. In the Atari benchmark, ROSMO outperforms offline RL baseline algorithms.


**Summary Of The Review:**

The paper provides modifications to MuZero Unplugged for offline RL, which makes the algorithm simpler yet more effective and this is a nice contribution. Still, the technical novelty is a bit limited, and it would be great to have more offline RL baselines in the experiments.

---

> ### Author Response · Authors · 2022-11-18
> **Response to Reviewer Dxeb (2/n)**
>
> > **Q4**: In Figure 4d, it seems OneStep is not yet converged. It would be great to provide the result with a longer number of steps to see whether ROSMO clearly outperforms OneStep at convergence. In the current result of Figure 4d, the conclusion is unclear to me.
>
> **A4**: Yes it is true that OneStep in Figure 4(d) on page 9 has not converged. We increased the training time as suggested and included more games to clarify the comparison. We have added the additional results in **Appendix F.2 on page 20**. As a quick summary, with well-covered data, ROSMO and OneStep reach similar final performances, while **ROSMO can learn faster**; when the data coverage is limited, **ROSMO significantly outperforms OneStep**.
>
>
> ***
> > **Q5**: It would be great to include the result of baseline algorithms in Figure 2. Why is ROSMO only being compared with MZU?
>
> **A5**: We did not include other baselines for BSuite because it was used to analyze and compare ROSMO with MZU for its cheaper computation (especially for running MZU). We have added the comparison of different methods on BSuite **in Table 2 on page 8**. The results show that **ROSMO achieves the highest performance for two out of three environments**, **after adding the additional baselines**.
>
>
> ***
> > **Q6**: In section 4.1.(3), why is MZU more sensitive to the parametrization of dynamics? It would be great to provide some possible reasons.
>
> **A6**: MZU relies on the Monte-Carlo Tree Search to construct the improvement targets. Accurately learned models are necessary for the constructed targets to be correct instead of misleading. With the parameterization of different capacities, we could expect the learned dynamics to have different levels of accuracy. When using these models, ROSMO could suffer less than MZU because it uses only a one-step model unroll instead of deep search, in addition to the behavior regularization, which constrains the policy not going too far from the behavior cloning policy. This could be why ROSMO can perform more robustly than MZU with different parametrization.
>
>
> ***
>
> We hope the answers have **addressed your major concerns and will change your evaluation of our paper**. If you have any further questions, please let us know.

---

> ### Author Response · Authors · 2022-11-18
> **Response to Reviewer Dxeb (1/n)**
>
> We thank the reviewer for providing insightful feedback and suggestions to help us make the paper more complete. With the revision, we address your questions below.
>
>
>
> ***
> > **Q1**: I am curious to see the result when only (11) is used for the policy improvement while not using (9), since it seems (11) can also serve as a policy improvement.
>
> **A1**: Only (Eq. 11) being applied for the policy improvement gives the “Behavior” variant, which we compare in **Figure 4(d) on page 9**. From the figure, we can see that “Behavior” is inferior to ROSMO due to the lack of policy improvement (Eq. 9). Admittedly, the filtered behavior cloning in behavior regularization is similar to the binary CRR form, with the advantage computed using the one-step model unroll (instead of learning Q and computing  $adv = Q - \mathop{\mathbb{E}}[Q])$). Hence we also compare binary CRR with the same network architecture and follow their hyperparameters from the official codes in Figure 4(d). The result shows that “Behavior” learns similarly to CRR with slightly better performance. The difference could be that how we formulate the advantage estimation (from value and reward prediction) is different from that of CRR (only using the model-free Q-value prediction).
>
>
> ***
> > **Q2**:  There are missing comparisons with model-based offline RL algorithms in the experiments.
>
> **A2**: We compare our method with MuZero Unplugged, regarded as the state-of-the-art offline algorithm for the RL Unplugged Atari benchmark. The other methods (e.g., MOPO [1], MOReL [2]) only focused on low-dimensional state-based domains. Therefore, it is unclear how their performance can transfer to image-based tasks. Though COMBO [3] extended MOPO and conducted experiments on image-based tasks, there are no reproducible codes nor the datasets they used available. We have spent a lot of effort (**four human-week + 16 A100 machines**) in implementing MOReL and COMBO and tested them in the Atari MsPacman environment (which is a representative game in Atari benchmarks, as explained in the paper). First, we adapted DreamerV2 for the world model, followed by the algorithms developed in MOReL and COMBO. The implementation details and results of MoReL and COMBO can be found in **Appendix F.1 on page 18**. We also attached all our codes (not well-cleaned yet working) for our original experiments and the newly added MOReL and COMBO experiments.
>
>
> As a summary, we observe that among all compared methods (i.e., **MOReL**, **COMBO**, MuZero Unplugged, and ROSMO), ROSMO (**ours**) **achieves the best result**. Due to time and resource constraints, we only provide the results on MsPacman for this rebuttal round. We will extend the experiments to more games to validate this conclusion and update the results later. Additionally, we note that the two-stage ensemble-based methods (i.e., MOReL, COMBO) require even heavier computation for image-based tasks. For a single game, one must first train *multiple* dynamics models, then train the RL agent using the ensemble of the models (which also poses challenges to GPU memory). This might hinder the practicability of such methods. Model exploitation is also observed during training of **MOReL** and **COMBO**, where the learned model return continues to grow while the evaluation return starts to decrease.
>
> [1] Yu, T., Thomas, G., Yu, L., Ermon, S., Zou, J. Y., Levine, S., Finn, C. & Ma, T. (2020). MOPO: Model-based offline policy optimization. Advances in Neural Information Processing Systems, 33, 14129-14142.
>
> [2] Kidambi, R., Rajeswaran, A., Netrapalli, P., & Joachims, T. (2020). MOReL: Model-based offline reinforcement learning. Advances in neural information processing systems, 33, 21810-21823.
>
> [3] Yu, T., Kumar, A., Rafailov, R., Rajeswaran, A., Levine, S., & Finn, C. (2021). COMBO: Conservative offline model-based policy optimization. Advances in neural information processing systems, 34, 28954-28967.
>
> ***
> > **Q3**:  In (1), there is no explicit definition of each term, $\ell^r, \ell^v, \ell^p$.
>
> **A3**: Added explanations for each loss term and highlighted them in blue on page 3. They refer to the reward, value, and policy loss functions. For example, the reward and value loss function could be a mean square error (MSE) or cross-entropy (CE) (for distributional agent); and the policy loss function is CE. In our case, we use CE for all of them.

---

> ### Author Response · Authors · 2022-11-24
> **any further questions?**
>
> Dear Reviewer Dxeb,
>
> We have added the suggested baseline algorithms, including MOReL and COMBO, into our revision. Our method ROSMO outperforms the newly added baselines.
>
> We also ran the experiments longer to demonstrate the advantages of the ROSMO algorithm compared to one of its improvement operators.
>
> We are wondering whether you have further concerns/questions so we can elaborate more. If not, could you please consider raising your evaluation score?
>
> Thanks,
> Authors

---

### Decision · Program_Chairs · 2023-01-20

**Decision:**

Accept: poster

**Justification For Why Not Higher Score:**

The method is novel and practical but uses existing methodological elements.

**Justification For Why Not Lower Score:**

The method provides great empirical performance on realistic datasets.

**Metareview: Summary, Strengths And Weaknesses:**

The authors present a novel method for model based off-policy RL that instead of performing monte-carlo-tree-search to identify a good continuation value of each action, it only performs a one-step advantage learning. The benefits are computational, but also in terms of robustness to model mis-specification and noisy dynamics. They show that their method outperforms other methods on the BSuite benchmark suite.

There is valid criticism that the elements invoked in this new method are not novel and that there was not sufficient comparison with other model-based approaches in the experiments, and only with mu-zero. However, the majority of reviewers value the practical aspect of the contribution and combining existing elements into a practically working algorithm is non-trivial.

Moreover, the reviewers have offered extensive responses to the reviewer criticism and have updated their manuscript with new experiments comparing to the aforementioned model-based approaches that were not there in the original submission; showing superior performance. Moreover, there was some criticism on the benchmark suite being used (compared e.g. to DR4L), to which the authors again provided satisfactory response. Therefore the main concerns have been addressed in the revision

**Note From Pc:**

if the above contains the word "oral" or "spotlight" please see: "oral" presentation means -> notable-top-5% and "spotlight" means -> notable-top-25%. As stated in our emails, we are disassociating presentation type from AC recommendations